# Temperature-Dependent Functional Response of *Harmonia axyridis* (Coleoptera: Coccinellidae) on the Eggs of *Spodoptera litura* (Lepidoptera: Noctuidae) in Laboratory

**DOI:** 10.3390/insects11090583

**Published:** 2020-09-01

**Authors:** Yasir Islam, Farhan Mahmood Shah, M. Abas Shah, Muhammad Musa Khan, Muhammad Asim Rasheed, Shakeel Ur Rehman, Shahzaib Ali, Xingmiao Zhou

**Affiliations:** 1Hubei Insect Resources Utilization and Sustainable Pest Management Key Laboratory, College of Plant Science and Technology, Huazhong Agricultural University, Wuhan 430070, China; yasirislam2143@gmail.com (Y.I.); asimrasheed@webmail.hzau.edu.cn (M.A.R.); shakeel.entomologist@hotmail.com (S.U.R.); shahzaibali@webmail.hzau.edu.cn (S.A.); 2Department of Entomology, Faculty of Agricultural Sciences and Technology, Bahauddin Zakariya University, Multan 60000, Pakistan; farhanshah0009@yahoo.com; 3ICAR-Central Potato Research Institute-Regional Station, Jalandhar Punjab-144003, India; mabas.shah@icar.gov.in; 4Key Laboratory of Bio-Pesticide Innovation and Application, South China Agricultural University, Guangzhou 510642, China; drmusakhan@outlook.com; 5Engineering Research Center of Biocontrol, Ministry of Education Guangdong Province, Guangzhou 510640, China

**Keywords:** attack rate, biological control, coccinellids, handling time, lepidoptera, predation

## Abstract

**Simple Summary:**

*Spodoptera litura* (Fabricius) (Lepidoptera: Noctuidae) is a notorious and polyphagous pest of several economically important agricultural crops. It is worldwide in distribution and primarily managed through typical dependence on insecticides, which resulted in health and the environmental challenges and selected for resistance development in *S. litura* field populations. Resistance caused chemical control failures and *S. litura* outbreaks around the world. This necessitated development of eco-friendly alternative approaches such as biological control. With this view, current study investigated the functional response of *Harmonia axyridis* (Pallas) (Coleoptera: Coccinellidae) at various growth stages (i.e., 1st, 2nd, 3rd and 4th instars, and male and female stages) and temperatures (i.e., 15, 20, 25, 30 and 35 °C) against *S. litura* eggs to enable the recognition of efficient biocontrol stages that could be utilized to suppress *S. litura* populations. In our findings, egg consumption depended on the growth stage of the predator as well as temperature. All stages consumed *S. litura* eggs, but more promising stages with active egg consumption were the 4th instar and adults (male and female) typically at higher temperatures (25–35 °C). We conclude that these stages may be exploited to suppress *S. litura* populations in fields and greenhouses.

**Abstract:**

*Spodoptera litura* (Fabricius) (Lepidoptera: Noctuidae) is a major pest of several economically important crops with worldwide distribution. Use of insecticides is the principal strategy for its management, which has subsequently led to insecticide resistance and control failures. Functional response of *Harmonia axyridis* (Pallas) (Coleoptera: Coccinellidae) at larval and adult stages was evaluated in this study, using *S. litura* eggs as the prey at various temperatures varying between 15 and 35 °C. Based on logistic model findings, linear parameters of various predatory stages of *H. axyridis* at various temperatures were significantly negative, which indicate a type II functional response. The theoretical maximum number (*T*/*T_h_*) of eggs consumed increased with increasing temperature across all predatory stages. According to the random predator equation, the coefficients of attack rate increased and that of handling time decreased as the temperature increased. The 4th instar and adult stages were superior candidates for biocontrol of the target prey, typically at higher temperatures. The maximum attack rate (0.546 ± 0.058 h^−1^) and lowest handling time (0.189 ± 0.004 h) were exhibited by the females at 30 and 35 °C, respectively, whereas these parameters were inferior for early instars. These findings clearly depict that the 4th instar and adult predators are efficient egg consumers and can serve as potential suppressors of *S. litura* field populations. The limitations of the predictions formulated by functional response trials are also discussed.

## 1. Introduction

The tobacco cutworm, *Spodoptera litura* (Fabricius) (Lepidoptera: Noctuidae) is well-known as a notorious cosmopolitan pest insect with extensive host range of economically important crops such as mallorca cabbage, *Brassica balearica* L. (Brassicales: Brassicaceae), maize, *Zea mays* L., rice, *Oryza sativa* L. (Poales: Poaceae), cotton, *Gossypium hirsutum* L., jute, *Corchorus olitorius* L., okra, *Abelmoschus esculentus* L. (Malvales: Malvaceae), tobacco, *Nicotiana tabacum* L., potato, *Solanum tuberosum* L., capsicum, *Capsicum annuum* L., tomato, *Solanum lycopersicum* L., eggplant, *Solanum melongena* L. (Solanales: Solanaceae), tea, *Camellia sinensis* L. (Ericales: Theaceae), soybean, *Glycine max* L. and peanut, *Arachis hypogaea* L. (Fabales: Fabaceae) [1]. It is a polyphagous pest of about 27 host species belonging to 25 genera of 14 families including cultivated crops, vegetables, weeds, fruits and ornamental plants [2,3]. High reproductive rate, rapid damage potential and extensive host range have contributed towards the economic status of the pest. A single female can lay over 2000 eggs during her entire life span of one week [2]. The early caterpillars are gregarious leaf eaters [4], become solitary as growth advances with potential to feed on the reproductive and fruit structures [5] of the attacked plant. Larval feeding can result in crop losses of 25.8% to 100% depending upon the susceptibility of the attacked host [6,7,8]. To protect these losses, use of insecticides is the primary strategy against *S. litura* in almost all agricultural areas of the world [9].

The control of agricultural pest challenges has depended mostly on the application of synthetic pesticides [10]. About 80% of the pesticides are used by the developed world and remaining is by the developing world [11]. In last years, pesticide use has considerably increased in China [12,13,14], corresponding to over 60% of the pesticide applied worldwide [11]. Resistance evolution is a major challenge associated with pesticides among other challenges for health and the environment [15,16]. *Spodoptera litura* has been shown to be resistant to a range of chemicals, and is ranked among the top ten most resistant insect pests of the world, with 638 reported case of resistance against 39 active ingredients [17]. Resistance to insecticides from synthetic origin including conventional and new chemistry chemicals has been reported in *S. litura* populations from Pakistan, India, China and United States [18,19]. Resistance development led to outbreaks by *S. litura* and failures of the crop around the world [3], particularly in Asia including India, Pakistan [20,21] and China [22]. Hence, the management of this pest has become increasingly difficult all over the world. The commonly used chemicals are ineffective in controlling the pest owing to resistance development and exploration of efficient and environmentally benign approaches such as biocontrol agents are desirable.

Biological control agents are considered the major forces regulating insect populations in global agroecosystems [23,24]. In particular, the coccinellids ladybeetles are an important group of insect predators, as they have large body size and prey upon multiple prey resources [25]. The assessment regarding natural enemy potential to suppress pest population is mainly carried out via evaluation of functional response, which depicts the interaction between prey density and predation capacity [26,27,28], enabling suitable choice of predator to be utilized in classical and augmentative biological control programs [29,30]. However, functional response may be regulated by a range of factors such as host plant of the prey [31,32,33,34,35,36], feeding behaviour and history [37], prey species [38,39], predator and its phenology [40,41] and temperature also [42,43,44,45]. In these, temperature is regarded as one of the most fundamental aspect, with ability to affect predator growth, development and foraging behaviour and thus the functional response [44].

The multicolored Asian ladybird beetle, *Harmonia axyridis* (Pallas) (Coleoptera: Coccinellidae) is a voracious predator that preys upon soft bodied hemipterans [46,47] and early-instar caterpillars [48]. Predation is also reported on the eggs of Monarch butterfly [49]. Many prior researches exploited *H. axyridis* to evaluate functional response against many crop pests [50,51,52,53], but information is lacking in *S. litura*. The current study aimed to assess the functional response of various stages of *H. axyridis* at five different temperatures using the eggs of *S. litura* as the prey. The information generated from the findings would assist to strengthen the biological control of *S. litura* under field conditions.

## 2. Materials and Methods

### 2.1. Insect Cultures

#### 2.1.1. Mass Rearing of *H. axyridis*

In January 2019, colonies of *Harmonia axyridis* and *Acyrthosiphon pisum* (Harris) (Hemiptera: Aphididae) were established from a population of adults taken from a stock culture at Key Laboratory of Hubei, Insect Resources Utilization and Sustainable Pest Management, located at Huazhong Agricultural University (HZAU), Wuhan, China. *Harmonia axyridis* was fed with *A. pisum* and both colonies were maintained under constant laboratory conditions at 23 ± 1 °C, 70 ± 5% RH and 16:8 (L:D) h photoperiod. The aphid species, *A. pisum* was reared on bean plants *Phaseolus vulgaris* L. (Fabales: Fabaceae). A total of 20 pairs of adult (male: female) were released in mesh covered cages (60 × 44 × 34 cm), containing bean plants heavily infested with aphids. Aphid culture was refreshed daily and plants were carefully observed for *H. axyridis* egg batches. Eggs batches were removed and shifted to a controlled conditioned growth chamber, maintained at 23 ± 1 °C, 70 ± 5% RH and 16:8 (L:D) photoperiod, in a Petri dish (6 cm) with tissue paper lined at the bottom. Eclosed larvae were removed and secluded in another Petri dish along with aphid prey.

#### 2.1.2. Mass Rearing of *S. litura*

*Spodoptera litura* population was established from a collection of 50 pupae obtained from, The Institute of Plant Protection and Soil Fertility Hubei Academy of Agricultural Sciences, Wuhan, China in 2019. All pupae were kept in a circular glass transparent jar (21 × 9 cm diameter; Model No. GG-17, manufactured by Sichuan Shubo Group Co., Ltd., Chongzhou, China) in an incubator (Shanghai Xinmiao Medical Device Manufacturing Co., Ltd., Shanghai, China: Model No QHX-250 BSH-III) at 27 ± 2 °C and 70 ± 5% RH with a photoperiod of 14:10 h (L:D). Emerged moths were collected and reared in another similar sized circular glass transparent jar (21 × 9 cm diameter) carrying a piece of cotton swab dipped in 20% honey solution as their diet. The jar was lined with tissue papers at the bottom as an egg laying substrate and the lid had multiple small holes for aeration. Once egg lying started, laid batches were removed daily and transferred to a small plastic box along with artificial diet [54] for eclosing larvae. The artificial diet (semi-solid) was consisted of kidney bean flour (150 g), yeast powder (24 g), methyl-4-hydroxy benzoate (1.5 g), ascorbic acid (2.35 g), distilled water 550 mL, agar (8.4 g), sorbic acid (0.75 g), streptomycin (0.75 g), and formaldehyde solution (1 mL) [55]. Feces were removed daily from the jar and the food was changed every 24 h. Larvae were reared in these boxes since they reached the final instar (6th instar), afterward shifted to another plastic box (12 × 7.5 cm diameter, purchased from Wenling Daxi Lingping Plastic Products Factory, Wenling, China) carrying diet and arrangement for pupation. This protocol was followed for developing and maintaining adult colonies.

### 2.2. Functional Response

For functional response studies, *H. axyridis* adults from the stock culture were shifted into similar sized plastic containers of 27 cm length, 14 cm width and 18 cm height. In total, twelve sets of plastic containers were prepared. Adults *H. axyridis* were adapted to prey on *S. litura* eggs. For this purpose, one pair of adult predator (male and female) was released in each container with 200 *S. litura* eggs refreshed every 12 h. A cotton wick was added to each container as an egg laying substrate. This whole apparatus was placed in an incubator under constant laboratory conditions at 23 ± 1 °C, 70 ± 5% RH and 16:8 (L:D) h photoperiod. The predator was reared on *S. litura* eggs for at least one complete generation, which ensured predator adaptation to successfully exploit eggs as prey. After this, functional response experiment was formally initiated. One fundamental aspect for the true assessment of predation capability is to maintain homogeneity of predator (i.e., instar/stage) age. To meet this crucial aspect, we kept male-female pairs together in a container carrying cotton wick as an egg lying substrate, checked, and collected every 12 h interval. Using this approach, different instars of approximately same age (0–6 h old after molting) were obtained. Upon getting each desired stage i.e., larval instars (first, second, third, and fourth) and adults (male and female), they were kept individually in a Petri dish of same size (6 cm). The first instar after eclosion was starved for about six hours, whereas subsequent instars/stages were starved for about 24 h within 24 h of their last molting. During starvation, only humidity was available in each petri dish by moist cotton roll. The experiments were performed in Petri dishes (6 cm diameter) on cotton leaves lined with fine layer of agar solution to prevent cotton (*Gossypium hirsutum* L. cv. Varamin) leaf discs (5 cm diameter) from drying during the experiment. A single fresh cotton leaf disc was centered upside down on the agar solution to obtain uniform leaf surface [56]. Freshly laid eggs were used for each replication. A fine camel hairbrush was used to transfer eggs into the Petri dish. First and second instar larvae were given 3, 6, 10, 15, 20, 25, 30, 35 and 5, 10, 15, 20, 30, 40, 50, 60 eggs, respectively, whereas third and fourth instar, and adult male and female were given same densities of eggs i.e., 5, 10, 25, 50, 100, 150, 250, 350 eggs. The temperature ranges considered reflect the thermal conditions experienced by predator in various protected and field crops in temperate areas. The experiment was conducted at five different constant temperatures (15 °C, 20 °C, 25 °C, 30 °C and 35 °C) with 70 ± 5% RH and photoperiod of 16:8 (L:D). The experiment was replicated 10 times simultaneously for each instar/stage, density and temperature. Eggs consumed/damaged by *H. axyridis* were not replaced during the entire experiment. After 24 h, the larval and adult predators were removed from the Petri dishes to count about the numbers of egg consumed.

### 2.3. Data Analysis

The Shapiro–Wilk test was used to assess the normality of data regarding mean and proportionate prey consumption by *H. axyridis* [57], prior to the analysis. The data were typically found to be non-normal (*p* < 0.05). The data on average prey consumption were analyzed with generalized linear model assuming negative binomial distribution due to over-dispersion and group means were separated with Tukey’s Honestly Significant Difference (HSD) test (*p* < 0.05). The data on proportionate prey consumption were subjected to the Kruskal–Wallis test (non-parametric ANOVA) for assessing significant effects, followed by Dunn’s multiple comparison test to differentiate between group means (*p* < 0.05). Egg consumption rate by predator at various predatory stages and temperatures was assessed separately.

A polynomial logistic regression equation was fitted for the proportion of prey eaten on the prey density offered, assuming a binomial distribution of data to determine the type of functional response [58] (Equation (1)).
(1)$$\frac{N_{e}}{N_{O}}=\frac{\exp\left(P_{o}+\;P_{1}N_{o}+\;P_{2}N_{o}^{2}+\;P_{3}N_{o}^{3}\right)}{1+\exp\left(P_{o}+\;P_{1}N_{o}+\;P_{2}N_{o}^{2}+\;P_{3}N_{o}^{3}\right)}$$
where *N_e_* and *N_o_* represent the number of prey consumed and initial prey density, respectively, and $\frac{Ne}{N0}$ is the proportion of prey consumed. The regression parameters *P*_0_, $\;P_{1}$, $\;P_{2}$, $\mathrm{and}\;P_{3}$ are the intercept, and the linear, quadratic and cubic coefficients, respectively. The coefficients were estimated by the maximum likelihood method. The signs of the linear and quadratic coefficients indicated the type of functional response. When *P*_1_ < 0, the functional response was type II, and if *P*_1_ > 0 and *P*_2_ < 0, the response was type III [58]. It is important to determine the nature of functional response for estimating the functional response parameters using an appropriate method. The type II response indicated that the prey consumption declined monotonically with prey density offered, and a type III response indicated that the proportionate prey consumption was positively density dependent [58]. The Roger’s random predator equation is appropriate for modelling predation or parasitism whenever predation or parasitism results in a significant reduction of prey or host densities [59] (Equation (2)).
(2)$$N_{e}=N_{o}\;\left[1-\exp\left(a\left(T_{h}N_{e\;}-T\right)\right)\right]$$
where $N_{e}$ and *N*_0_ represent the number of consumed and prey density offered, respectively, a is the instantaneous attack rate, $T_{h}$ is the handling time, *T* is the duration of the experiment (24 h). We used “*glm*” function to fit the logistic regression, and the “*friar*” [60] package to determine the coefficients of attack rate and handling time. All statistical analyses were done in R 4.0.0 [61].

The theoretical maximum predation rate, given by the ratio of *T*/*T_h_*, represents the maximal prey consumption in the given time interval. We calculated the maximum predation rate from the estimates of *T_h_* as determined above and subjected to non-parametric ANOVA [62].

## 3. Results

Generally, the mean prey consumption rate increased with increasing temperature. The mean prey consumption rate increased from 20.18 ± 6.25 to 43.70 ± 10.12 (mean ± SE) eggs per day per predator as the temperature increased from 15 to 35 °C. The mean prey consumption was lowest at 15 °C, intermediate at 20 and 25 °C and greatest at 30 and 35 °C (Table 1).

The predator consumed 42% of the offered prey at 15 °C, which increased to 75% at 35 °C (Table 2). Mean prey consumption rate increased with advancing predator growth stages. The 1st and 2nd instar consumed the lowest. The highest prey consumption was noted for final instar and adult stages. Among these stages, consumption rate was highest for adult females (57.39 ± 4.36) but similar to that by 4th instar (52.45 ± 6.66) and adult males (43.12 ± 6.00) (Table 1). The 1st instar predator could consume 45% of the prey offered while the adult females and the 4th instar consumed 75% and 72% of the prey offered, respectively in the same time interval (Table 2). When effects were assessed on various temperatures, 4th instar and adult predators (male and female) were superior egg consumers over the other growth stages.

Based on *S. litura* egg consumption rates, all stages of *H. axyridis* showed a type II functional response across all temperatures, as confirmed by linear estimates being significantly negative for all stages at all tested temperatures, which is an inverse density dependent response and the random predator equation is more appropriate for experimental data of type II functional response. The proportionate prey consumption rate was greater at low prey densities but lowered at higher prey densities. This type of functional response was further confirmed by negative linear estimates $(P_{1}$ < 0) according to logistic regression analysis (Table 3 and Figure 1).

The curves reflect that at initially low prey densities, egg consumption rates quickly rose across all predator stages at all tested temperatures but levelled off with a decreasing trend with beyond a certain increase in prey density (Figure 1).

The attack rate increased with increasing temperature across all predatory stages; however, significant effects were obtained for the initial instars (1st and 2nd) and for adult females only. For adult females, the attack rate increased from 0.155 ± 0.01 h^−1^ at 15 °C to 0.546 ± 0.05 h^−1^ at 35 °C. Similarly, the attack rate for fourth instar grubs increased from 0.194 ± 0.018 h^−1^ to 0.33 ± 0.02 h^−1^ from 15 to 35 °C (Table 4). Attack rate estimate was significantly greater at 30 and 35 °C as compared to lower temperatures. The highest attack rate was observed for females (0.546 ± 0.058 h^−1^) and males (0.313 ± 0.024 h^−1^) at 35 °C whereas for the 4th instar, highest attack rate (0.243 ± 0.016 h^−1^) was noted at 30 °C. Overall, the attack rates of the fourth instar and adult female were significantly superior over the rest of stages for which the attack rates did not vary significantly. Therefore, attack rate was probably not a good indicator of the predator efficiency for the predator-prey combination under study.

The handing time of predator decreased with increasing temperature, and with advancing in growth stages. The lowest handling times was exhibited by all predatory stages at 35 °C. For adult females, the handling time decreased from 0.322 ± 0.008 h at 15 °C to 0.199 ± 0.004 h at 35 °C. For the 4th instar, the handling time decreased from 0.55 ± 0.014 h at 15 °C to 0.174 ± 0.003 h at 35 °C (Table 5). At 15 °C, the highest handling time was noted for the 1st instar (11.71 ± 1.75 h) while the lowest handling time was noted for adult female (0.322 ± 0.008 h) followed by the 4th instar (0.551 ± 0.014 h). The female took significantly much less time to handle prey than the 4th instar at 15 and 20 °C whereas no significant difference was found at other temperature for these predatory stages. Overall, the handling time across all temperatures was similar for 1st, 2nd and 3rd instar grubs and significantly higher as compared to 4th instar and both adult stages.

The theoretical maximum predation rate (T/Th) significantly increased with advancing in growth stages, and with increasing temperature. For the 4th instar grub, the maximum predation rate increased from 43.64 ± 5.63 to 137.93 ± 19.45 as the temperature increased from 15 to 35 °C. Similarly, for adult females, the predation rate increased from 75.00 ± 11.10 to 120.60 ± 3.76 as the temperature increased from 15 to 35 °C. Among the predator growth stages, maximum predation rate was exhibited by the adult females followed by the 4th instar across all temperatures with the exception of 35 °C (Figure 2). The average maximum predation rate (averaged across all temperatures) was highest for adult females (108.52 ± 9.53), followed by 4th instars (96.31 ± 16.35) and adult males (72.94 ± 12.31); rest of growth stages performed comparatively lower.

## 4. Discussion

Functional response is an imperative criterion for assessing the efficiency of a predator on a given prey. Predation by *H. axyridis* using functional response experiments has been investigated against many crop pests [63,64]. However, no prior study reports predatory potential on *S. litura* eggs. As of today, current study assessed the predatory efficiency of *H. axyridis*, at various predatory stages, against *S. litura* eggs at a range of temperature that are likely to be experienced by the predator under field conditions.

Our findings showed that all developmental stages of *H. axyridis* across all tested temperatures exhibited a type II functional response. This type II functional response has been commonly reported for coccinellids [40,65,66] including *H. axyridis* when fed on various prey species such as *Rhopalosiphum prunifoliae* L. [50], *Cinara* sp. [51], *Aphis craccivora* K. [67], *Schizaphis graminum* R. [68], *Lipaphis erysimi* K. [52], *Myzus persicae* S. [69], *Myzus nicotianae* S. [70], *Rhopalosiphum nymphaeae* L. [50], *Melanaphis sacchari* Z. [63] (Hemiptera: Aphididae) and eggs of *Danaus plexippus* L. (Lepidoptera: Nymphlidae) [49]. In our findings, temperature affected the magnitude of the egg predation by various stages of predator. Increased temperature increased the numbers of eggs eaten. Higher consumption at higher temperature could be attributed to the faster growth and development. Temperature-related changes affect predator biology and physiology, such as higher reproduction, metabolism, and activity of predator and are likely to increase with increasing temperature [71,72]. High temperature increased the functional response of *H. axyridis* by decreasing handling time and increasing attack rate, in accord with many previous researches [73,74,75,76,77]. These findings suggest regulatory role of temperature towards searching efficiency and handling time across all predatory stages [64].

Handling time and attack rate (also called search rate or searching time or rate of discovery or space clearance rate) are important parameters of the functional response. The attack rate determines the ability of a predator to catch prey in a given time, and handling time indicates the time a predator spends to identify, subjugate, attack and consume a particular prey [1]. These parameters determine the foraging behavior of a predator and explain how much efficient a predator is against a given density of prey. As the foraging behavior increases, the attack rate will also increase which leads to a decrease in handling time. It means the predator will consume more prey and we can use it to control pest. The parameters of functional response help to explain relative changes in the efficacy of the predator with changing density of the prey, along the growth of the predator. For example, early in the season when the pest density is low, the predator efficiency is also expected to be low, based on the parameters of functional response. Similarly, as the predator grows through different instars, the functional response parameters also improve, which means that the capacity of the predator to regulate the pest population will improve. Therefore, the functional response parameters give an indication of the temporal variation in the efficacy of the predator to regulate the pest population, which could be exploited to improve the fate of the control program involving agents of biological control.

Our findings showed that both the handling time and attack rate depend on temperature as well as size of the predator. The attack rate for first instars of *H. axyridis* was very low compared to other stages at all tested temperatures, in line with findings of another study where this predator was fed with aphid species [64]. We found very low attack rate for first instar among all stages i.e., 0.03 ± 0.016 h^−1^, and these findings are supported by another study where attack rate for first instar i.e., 0.003 ± 0.006 h^−1^ was also very low when *H. axyridis* was fed with *A. gossypii* [64]. The low attack rate might be due to smaller sizes, slower movements and low nutrition requirement of the early instar. The larvae in second and third instars had relatively higher *S. litura eggs* consumption compared to the first instar larvae. Moreover, fourth instars larvae and female adults of *H. axyridis* were more voracious than other developmental stages, likely due to nutritional requirements for development and for reproduction [49]. The highest attack rate was exhibited by females i.e., 0.546 ± 0.058 h^−1^ at 35 °C. Similarly, high attack rate was observed in female i.e., 0.94 ± 0.374 h^−1^ and 0.78 h^−1^ when *H. axyridis* allowed to prey on *D. plexippus* eggs and *Aphis glycines* Matsumura (Hemiptera: Aphididae) respectively [49,78]. Likewise, fourth instar and female of *Hippodamia variegata* G. (Coleoptera: Coccinellidae) showed highest attack rate against *A. craccivora* i.e., 0.144 ± 0.013 h^−1^ and 0.106 ± 0.011 h^−1^ respectively [36]. Literature suggests that the attack rate varies from predator species and stage, arena size, temperature and prey stage also [79]. Fourth instar larvae and females generally had higher search rates than third instar larvae, possibly reflecting increased mobility skills with developmental stage. The same results were observed when *H. axyridis* was fed on the aphid *M. persicae* [80]. Fourth instar is a more voracious feeder compared to other immature stages because of higher food and energy demands in order to grow and achieve the desired weight for pupation [81].

Handling time is an accurate measure for consumption rate and to determine the productivity of a predator as it represents the combined effect of time used to catch, kill, and digesting the prey [82]. In case of our study, handling time decreased with increasing temperature and by advancing in predator size also, indicating that these stages would more efficiently contribute in handling prey and typically at higher temperature. Handling time for fourth instar (0.174 ± 0.003 h) was shorter than adult female (0.189 ± 0.004 h) and male (0.243 ± 0.05 h). Better efficacy by fourth instar and adult predators over earlier instars, as found in our study, has been supported by findings of many earlier studies where a coccinellid, *Nephus arcuatus* K. (Coleoptera: Coccinellidae) was fed on spherical mealybug, *Nipaecoccus viridis* N. (Hemiptera: Pseudococcidae) [83], and also when *H. axyridis* was fed on aphid, *M. persicae* [80] and when *H. variegata* was fed on *Aphis fabae* S. [40]. Seko and Miura [80] reported lower handling time for female (0.127 h) followed by the male (0.146 h) and fourth instar (0.156 h) *H. axyridis*, whereas Farhadi, Allahyari and Juliano [40] reported lower handling time for female (0.409 ± 0.048 h) followed by fourth instar (0.454 ± 0.028 h) and male (1.194 ± 0.069 h) *H. variegate.* This also suggests that handling time is likely to be different according to prey and predator species, their phenology as well as physical nature of the prey, either static or mobile [84].

In general, predators spent more time handling preys at lower temperatures than at higher temperatures [44]. Ladybugs feeding on lepidopterans pests showed higher space clearance rates compared to the other pests. This is due to the high success of the attack rate, because all lepidopteran preys were relatively easy to subdue at egg or first instar stage [85,86,87]. Although our findings demonstrate that predators have increased space clearance rates at higher temperatures indicating that low-density prey populations in warmer environments can be reduced most successfully, many individuals have high feeding rate at intermediate temperature [88]. This strong temperature influence on functional response parameters can lead to important changes in predator-prey relationship, population dynamics and food web interactions [89,90,91].

From the above findings, it is clear that 4th instar and adults of *H. axyridis* are the important biocontrol agents of *S. litura* eggs, with more consumption typically at higher temperature. Due to its high attack rate over a wide range of temperatures, particularly at high temperatures (25–35 °C), *H. axyridis* is a good natural enemy for use against *S. litura* eggs in warm locations such as greenhouses. Although, the type of functional response is considered to be a significant aspect, it is not the only standard measuring the failure or success of a natural enemy. Functional response experiments under small scale laboratories are very unlike to natural environment and therefore the outcomes should be explained correctly with due caution [92,93]. Moreover, biocontrol efficiency of this ladybug can be appreciated only when considering all relevant attributes of their biology, including their population growth parameters. The studies under lab may depict the predation capacity of *H. axyridis* when eggs are abundant, but they do not count cannibalism or emigration of the natural enemy that may occur when *S. litura* populations drop significantly, leaving few natural enemies in the field, which would favor resurgence of the pest. Furthermore, it is paramount to consider prey development with respect to temperature when investigating functional response. As increasing temperature can also accelerate egg hatching, it is likely that predation may decrease with plenty of prey availability at higher temperatures. It thus worth to account for predator as well as prey biology with respect to temperature to synthesize careful conclusions. Therefore, studies performed under greenhouse or field conditions are required to comprehend the feeding behavior of *H. axyridis* in various cropping systems, in order to design practical release approaches for this promising coccinellids predator for natural suppression of *S. litura* field populations.

## 5. Conclusions

This study concluded that *H. axyridis* is an efficient *S. litura* egg consumer. The biocontrol efficacy of the predator is growth stage and temperature dependent. All stages of the predator feed on eggs but 4th instar and adults are the most efficient consumers than other stages, and predation is typically more active at higher temperatures (25–35 °C). We can use 4th instar and adult predators in greenhouses and fields to control this notorious pest.

## Figures and Tables

**Figure 1 insects-11-00583-f001:**
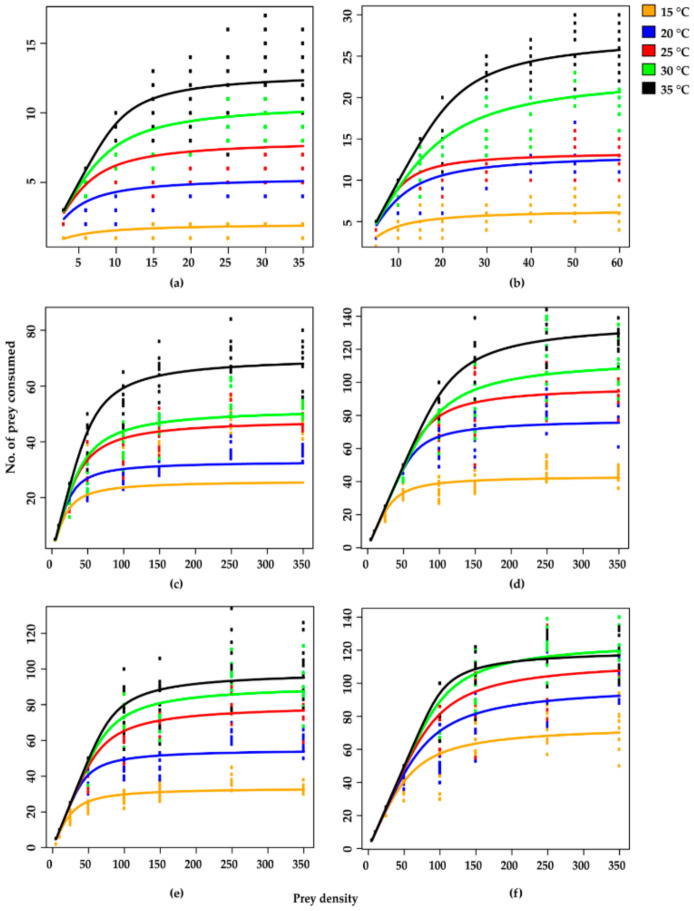
Type II functional response curves fitted by Roger’s random predator equation (RRPE) of different stages of *Harmonia axyridis* against *Spodoptera litura* eggs at various temperatures (**a**–**f**). Here, (**a**) 1st instar, (**b**) 2nd instar, (**c**) 3rd instar, (**d**) 4th instar, (**e**) adult male and (**f**) female.

**Figure 2 insects-11-00583-f002:**
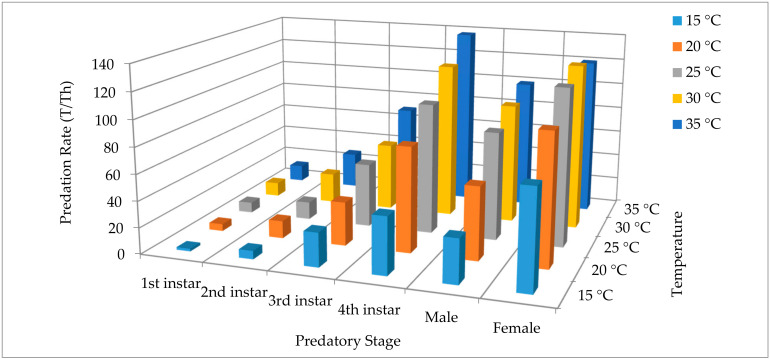
Comparison of the theoretical maximum predation rates for all the predatory stages of *Harmonia axyridis* on *Spodoptera litura* eggs at five different temperatures.

**Table 1 insects-11-00583-t001:** Mean prey consumption (±SE) rate on *Spodoptera litura* eggs by *Harmonia axyridis* at various predatory stages and temperatures.

Predator Stage	Temperature					Mean
	15 °C	20 °C	25 °C	30 °C	35 °C	
1st instar	1.61 ± 0.12 a/A	4.39 ± 0.37 a/B	6.35 ± 0.62 a/BC	8.04 ± 0.94 a/C	9.78 ± 1.31 a/C	6.03 ± 1.42
2nd instar	5.23 ± 0.39 b/A	10.14 ± 1.00 ab/B	11.08 ± 1.12 a/B	14.95 ± 2.07 ab/BC	18.34 ± 2.79 ab/C	11.95 ± 2.23
3rd instar *	18.89 ± 3.19 c/*	23.78 ± 4.19 bc/*	31.21 ± 6.28 b/*	33.03 ± 6.83 bc/*	43.05 ± 9.50 bc/*	29.99 ± 4.14
4th instar *	29.50 ± 5.57 cd/*	47.69 ± 10.93 c/*	56.24 ± 13.90 b/*	60.11 ± 15.45 c/*	68.71 ± 18.49 c/*	52.45 ± 6.66
Male *	23.16 ± 4.07 cd/*	36.56 ± 7.57 c/*	47.10 ± 10.81 b/*	52.18 ± 12.52 c/*	56.58 ± 14.06 c/*	43.12 ± 6.00
Female *	42.70 ± 9.90 d/*	52.91 ± 13.18 c/*	59.95 ± 15.39 b/*	65.60 ± 17.46 c/*	65.78 ± 17.21 c/*	57.39 ± 4.36
**Mean**	20.18 ± 6.25	29.24 ± 8.10	35.32 ± 9.36	38.98 ± 9.83	43.70 ± 10.12	

Means within the same column followed by the same letter (lower case) are not significantly different (Tukey’s HSD Test, *p* < 0.05). Means within the same row followed by the same letter (upper case) are not significantly different (Tukey’s HSD Test, *p* < 0.05). * Effect of temperature is non-significant (*p* > 0.05).

**Table 2 insects-11-00583-t002:** Proportionate prey consumption (±SE) rate on *Spodoptera litura* eggs by *Harmonia axyridis* at various predatory stages and temperatures.

Predator Stage	Temperature					Mean
	15 °C	20 °C	25 °C	30 °C	35 °C	
1st instar	0.14 ± 0.03 a/A	0.36 ± 0.08 */AB	0.49 ± 0.10 b/B	0.59 ± 0.09 b/B	0.68 ± 0.09 b/B	0.45 ± 0.10
2nd instar	0.28 ± 0.07 ab/A	0.50 ± 0.09 */AB	0.55 ± 0.10 ab/C	0.66 ± 0.08 ab/C	0.77 ± 0.08 b/C	0.55 ± 0.08
3rd instar	0.43 ± 0.13 ab/*	0.50 ± 0.13 */*	0.56 ± 0.13 */*	0.58 ± 0.13 */*	0.67 ± 0.12 */*	0.55 ± 0.04
4th instar	0.56 ± 0.13 b/*	0.70 ± 0.12 */*	0.75 ± 0.11 */*	0.77 ± 0.10 */*	0.82 ± 0.09 */*	0.72 ± 0.04
Male	0.48 ± 0.13 b/*	0.62 ± 0.13 */*	0.69 ± 0.11 */*	0.73 ± 0.11 */*	0.75 ± 0.11 */*	0.65 ± 0.05
Female	0.65 ± 0.12 b/*	0.72 ± 0.11 */*	0.77 ± 0.10 */*	0.80 ± 0.09 */*	0.81 ± 0.10 */*	0.75 ± 0.03
**Mean**	0.42 ± 0.08	0.57 ± 0.06	0.64 ± 0.05	0.69 ± 0.04	0.75 ± 0.03	

Means within the same column followed by the same letter (lower case) are not significantly different (Dunn’s Multiple Comparison Test, *p* < 0.05). Means within the same row followed by the same letter (upper case) are not significantly different (Dunn’s Multiple Comparison Test, *p* < 0.05). * Effect of temperature is non-significant (Kruskal-Wallis Test, *p* > 0.05).

**Table 3 insects-11-00583-t003:** Maximum likelihood estimates from logistic regression analyses of the proportion of prey eaten by different stages of *Harmonia axyridis* against initial number of *Spodoptera litura* eggs offered.

Temperatures	Growth Stages	Parameters	Estimates	S.E.	*Z*-Value	*Pr* (z)
15 °C	1st instar	Intercept	−0.074	0.60	−0.124	0.9017
		Linear	−0.242	0.12	−1.929	0.0537
	2nd instar	Intercept	1.47	4.10 × 10^−01^	3.595	<0.05
		Linear	−2.08 × 10^−01^	4.73 × 10^−02^	−4.405	1.06 × 10^−05^
	3rd instar	Intercept	1.61	1.38 × 10^−01^	11.67	<2 × 10^−16^
		Linear	−4.76 × 10^−02^	3.13 × 10^−03^	−15.20	<2 × 10^−16^
	4th instar	Intercept	3.21	1.77 × 10^−01^	18.10	<2 × 10^−16^
		Linear	−5.98 × 10^−02^	3.48 × 10^−03^	−17.18	<2 × 10^−16^
	Male	Intercept	2.09	1.46 × 10^−01^	14.32	<2 × 10^−16^
		Linear	−4.85 × 10^−02^	3.14 × 10^−03^	−15.44	<2 × 10^−16^
	Female	Intercept	3.59	2.01 × 10^−01^	17.86	<2 × 10^−16^
		Linear	−5.36 × 10^−02^	3.66 × 10^−03^	−14.64	<2 × 10^−16^
20 °C	1st instar	Intercept	2.40	5.65 × 10^−01^	4.256	<2 × 10^−16^
		Linear	−4.16 × 10^−01^	1.05 × 10^−01^	−3.964	<2 × 10^−16^
	2nd instar	Intercept	3.67	5.15 × 10^−01^	7.126	<2 × 10^−16^
		Linear	−2.92 × 10^−01^	5.29 × 10^−02^	−5.525	<2 × 10^−16^
	3rd instar	Intercept	2.34	1.51 × 10^−01^	15.46	<2 × 10^−16^
		Linear	−5.31 × 10^−02^	3.21 × 10^−03^	−16.54	<2 × 10^−16^
	4th instar	Intercept	6.54	3.65 × 10^−01^	17.91	<2 × 10^−16^
		Linear	−9.11 × 10^−02^	5.95 × 10^−03^	−15.30	<2 × 10^−16^
	Male	Intercept	4.43	2.27 × 10^−01^	19.51	<2 × 10^−16^
		Linear	−7.27 × 10^−02^	4.10 × 10^−03^	−17.74	<2 × 10^−16^
	Female	Intercept	3.59	2.01 × 10^−01^	17.86	<2 × 10^−16^
		Linear	−5.36 × 10^−02^	3.66 × 10^−03^	−14.64	<2 × 10^−16^
25 °C	1st instar	Intercept	4.00	7.15 × 10^−01^	5.604	2.10 × 10^−08^
		Linear	−5.27 × 10^−01^	1.20 × 10^−01^	−4.36	1.27 × 10^−05^
	2nd instar	Intercept	6.17	6.82 × 10^−01^	9.050	<2 × 10^−16^
		Linear	−5.05 × 10^−01^	6.59 × 10^−02^	−7.66	1.86 × 10^−14^
	3rd instar	Intercept	2.902	1.68 × 10^−01^	17.27	<2 × 10^−16^
		Linear	−5.40 × 10^−02^	3.31 × 10^−03^	−16.31	<2 × 10^−16^
	4th instar	Intercept	7.70	4.80 × 10^−01^	16.02	<2 × 10^−16^
		Linear	−9.82 × 10^−02^	7.50 × 10^−03^	−13.09	<2 × 10^−16^
	Male	Intercept	5.13	2.88 × 10^−01^	17.82	<2 × 10^−16^
		Linear	−6.92 × 10^−02^	4.85 × 10^−03^	−14.27	<2 × 10^−16^
	Female	Intercept	6.41	4.18 × 10^−01^	15.33	<2 × 10^−16^
		Linear	−7.48 × 10^−02^	6.61 × 10^−03^	−11.30	<2 × 10^−16^
30 °C	1st instar	Intercept	5.00	0.88	5.645	1.65 × 10^−08^
		Linear	−0.56	0.14	−4.05	5.00 × 10^−05^
	2nd instar	Intercept	4.76	6.68 × 10^−01^	7.123	1.05 × 10^−12^
		Linear	−2.91 × 10^−01^	6.35 × 10^−02^	−4.58	4.47 × 10^−06^
	3rd instar	Intercept	2.89	1.69 × 10^−01^	17.11	<2 × 10^−16^
		Linear	−5.15 × 10^−02^	3.29 × 10^−03^	−15.62	<2 × 10^−16^
	4th instar	Intercept	7.20	4.72 × 10^−01^	15.23	<2 × 10^−16^
		Linear	−8.65 × 10^−02^	7.37 × 10^−03^	−11.73	<2 × 10^−16^
	Male	Intercept	7.08	4.20 × 10^−01^	16.85	<2 × 10^−16^
		Linear	−9.33 × 10^−02^	6.69 × 10^−03^	−13.95	<2 × 10^−16^
	Female	Intercept	1.04 × 10^+01^	7.46 × 10^−01^	13.99	<2 × 10^−16^
		Linear	−1.27 × 10^−01^	1.10 × 10^−02^	−11.48	<2 × 10^−16^
35 °C	1st instar	Intercept	12.16	1.98	6.139	8.31 × 10^−10^
		Linear	−1.43	0.27	−5.30	1.16 × 10^−07^
	2nd instar	Intercept	9.87	1.46	6.75	1.46 × 10^−11^
		Linear	−5.76 × 10^−01^	1.19 × 10^−01^	−4.83	1.36 × 10^−06^
	3rd instar	Intercept	4.48	2.47 × 10^−01^	18.16	<2 × 10^−16^
		Linear	−6.30 × 10^−02^	4.30 × 10^−03^	−14.65	<2 × 10^−16^
	4th instar	Intercept	7.12	6.30 × 10^−01^	11.29	<2 × 10^−16^
		Linear	−6.27 × 10^−02^	9.51 × 10^−03^	−6.59	4.34 × 10^−11^
	Male	Intercept	8.29	5.16 × 10^−01^	16.06	<2 × 10^−16^
		Linear	−1.07 × 10^−01^	8.00 × 10^−03^	−13.39	<2 × 10^−16^
	Female	Intercept	1.27 × 10^+01^	1.03	12.34	<2 × 10^−16^
		Linear	−1.49 × 10^−01^	1.49 × 10^−02^	−10.01	<2 × 10^−16^

**Table 4 insects-11-00583-t004:** Coefficients of attack rate (mean ± SE) of *Harmonia axyridis* at various predatory stages and temperatures, preying upon *Spodoptera litura* eggs (according to Roger’s random predator equation).

Predator Stage	Temperature				
	15 °C	20 °C	25 °C	30 °C	35 °C
1st instar	0.03 ± 0.016 (0.022–0.039)	0.118 ± 0.035 (0.075–0.204)	0.177 ± 0.04 (0.127–0.252)	0.202 ± 0.039 (0.139–0.282)	0.373 ± 0.089 (0.255–0.727)
2nd instar	0.079 ± 0.022 (0.052–0.169)	0.156 ± 0.025 (0.114–0.236)	0.292 ± 0.06 (0.193–0.518)	0.163 ± 0.018 (0.125–0.214)	0.317 ± 0.04 (0.202–0.523)
3rd instar	0.12 ± 0.015 (0.089–0.172)	0.176 ± 0.02 (0.128–0.233)	0.155 ± 0.014 (0.112–0.225)	0.151 ± 0.013 (0.101–0.224)	0.233 ± 0.019 (0.164–0.330)
4th instar	0.194 ± 0.018 (0.146–0.281)	0.344 ± 0.033 (0.234–0.549)	0.351 ± 0.037 (0.236–0.506)	0.246 ± 0.016 (0.178–0.380)	0.333 ± 0.022 (0.232–0.544)
Male	0.133 ± 0.013 (0.097–0.182)	0.274 ± 0.03 (0.187–0.399)	0.24 ± 0.018 (0.160–0.409)	0.278 ± 0.021 (0.183–0.508)	0.354 ± 0.037 (0.235–0.544)
Female	0.155 ± 0.011 (0.117–0.215)	0.183 ± 0.012 (0.134–0.261)	0.243 ± 0.016 (0.176–0.372)	0.313 ± 0.024 (0.209–0.579)	0.546 ± 0.058 (0.270–2.912)

Values in parenthesis are 95% asymptote CIs. Means within columns are showing effect of temperature and within rows are showing effect of growth stage on attack rate.

**Table 5 insects-11-00583-t005:** Coefficients of handling time (mean± SE) of *Harmonia axyridis* at various predatory stages and temperatures, preying upon *Spodoptera litura* eggs (according to Rogers’ random predator equation).

Predator Stage	Temperature				
	15 °C	20 °C	25 °C	30 °C	35 °C
1st instar	11.715 ± 1.759 (10.99–12.96)	4.448 ± 0.337 (4.019–4.845)	2.972 ± 0.176 (2.741–3.246)	2.216 ± 0.121 (1.982–2.399)	1.85 ± 0.086 (1.691–2.008)
2nd instar	3.717 ± 0.276 (3.373–4.178)	1.806 ± 0.087 (1.680–1.946)	1.776 ± 0.073 (1.630–1.913)	1.036 ± 0.047 (0.942–1.129)	0.863 ± 0.031 (0.794–0.922)
3rd instar	0.92 ± 0.031 (0.867–0.996)	0.726 ± 0.021 (0.683–0.769)	0.498 ± 0.013 (0.462–0.538)	0.46 ± 0.012 (0.426–0.499)	0.339 ± 0.007 (0.322–0.359)
4th instar	0.551 ± 0.014 (0.520–0.584)	0.308 ± 0.006 (0.289–0.332)	0.244 ± 0.005 (0.224–0.267)	0.208 ± 0.004 (0.188–0.230)	0.174 ± 0.003 (0.161–0.187)
Male	0.715 ± 0.021 (0.678–0.761)	0.435 ± 0.01 (0.407–0.462)	0.299 ± 0.006 (0.280–0.319)	0.262 ± 0.005 (0.241–0.282)	0.243 ± 0.005 (0.223–0266)
Female	0.322 ± 0.008 (0.289–0.356)	0.242 ± 0.005 (0.222–0.270)	0.209 ± 0.004 (0.190–0.230)	0.189 ± 0.004 (0.175–0.205)	0.199 ± 0.004 (0.185–0.215)

Values in parenthesis are 95% asymptote CIs. Means within columns are showing effect of temperature and within rows are showing effect of growth stage on handling time.

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
