# Peer review of "Temperature-Dependent Functional Response of Harmonia axyridis (Coleoptera: Coccinellidae) on the Eggs of Spodoptera litura (Lepidoptera: Noctuidae) in Laboratory"

_insects, 2020, doi:10.3390/insects11090583_

Round 1
Reviewer 1 Report
This manuscript presents valuable information about temperature-dependent functional response of Harmonia axyridis (Coleoptera: Coccinellidae) on the eggs of Spodoptera litura (Lepidoptera: Noctuidae) under laboratory conditions. This is an important contribution being this the first documentation of the biological control of S. litura by predatory action of different instar larvae and adults of the Asian lady beetle.
Materials and Methods are very vague. That section needs major revisions and explanations.
Please find attached file with suggestions

Author Response
Dear Editor and Reviewer,
On behalf of all authors, I am thankful for valuable comments and suggestions on the manuscript insects-865851, entitled "Temperature dependent functional response of Harmonia axyridis (Coleoptera: Coccinellidae) on the eggs of Spodoptera litura (Lepidoptera: Noctuidae) in laboratory". These comments are valuable and helpful for revising and improving the manuscript, as well as this is an important guidance for our future research.
Reply to Comments of the editor
- Improvement to writing and data presentation
Reply: Thanks for your valuable suggestions and technical comments. These comments really helped us and improved the quality of our manuscript. We thoroughly re-checked all of our analysis, references and improved them according to your suggestions.
- Assess the effect of temperature on the prey as well, in adiition to the beahviour of the predator
Reply: The predaror and prey stock cultures were maintained at same condition (27 ± 2 °C and 70 ± 5 % RH with a photoperiod of 14:10 h (L:D) throughout the experiment. The eggs and one predator per petri plate were exposed to the desired remperature for the duration of the experiment (24 h). Since the actual evaluation time was short, therefore no appreciable effect is expected on the growth of the prey. Since we used eggs as prey, no change is expected in their behavior during the short period of the experinment.
And the responses to Reviewer 1's comments have been attached, check it please. Thank you!
-------------------------------------------
Reviewer 1
General Comments: This manuscript presents valuable information about temperature-dependent functional response of harmonia axyridis (Coleoptera: Coccinellidae) on the eggs of Spodoptera litura (Lepidoptera: Nocctuidae) under laboratory conditions. This is an important contribution being this the first documentation of the biological control of S. litura by predatory action of different instar larbvae and adults of Asian lady beetle.
Materials and Methods are very vague. That section needs major revisions and explanations.
Response: Thank you very much for considering our research valuable and important. We followed your all suggestions and revised our manuscript. It has improved the writing quality of our manuscript.
Minor Comments:
1) Line 23: Remove first sentence from Abstract.
Response: Thanks for technical comment. It has been removed from manuscript. Line number 23
2) Line 27: Remove here, and start from Functional
Response: Thanks for this valuable recommendation.It has been done in revised manuscript. Line number 26
3) Line 28: Replace “studied with “was evaluated in this study”
Response: Thanks for important comment. Have done in a revised manuscript. Line number 27
4) Line 37: the 4th instar??
Response: Here, early instars means first and second instar.
5) Line 46-47: Write Scientific Names, authority, order and family
Response: Thanks for this techanical suggestion. It has been added in revised manuscirpt. Line numbers 45-50
6) Line 66-69: too repetive and confusing
Response: Thanks. It has been revised in manuscript. Line number 70
7) Line 94-95: Replace “predator” with “colonies of Harmonia axyridis and acyrthosiphon pisum, Harris (Homoptera: Aphididae) ”. Replace “was” with “were”. Remove H. axyridis at. Replace “adult” with “adults”. Removed stage.
Response: Thanks for this valuable recommendation. Done in revised manuscript. Line number 97-98
8) Line 97: “Replace rearing of the predator was carried out” with “Harmonia axyridis was fed with”. Remove “maintained and replace it with “Both colonies were maintained”
Response: Done in revised manuscript according to valuable suggestion of reviewer. Line number 100-102
9) Line 98: Remove cyrthosiphon from…. Acyrthosiphon. Remove taken
Response: Done in revised manuscript. Line number 102
10) Line 99: Remove “from a laboratory”. Remove “Colony”
Response: Revised as suggested in manuscript. Line number 102
11) Line 99: bean plant Scientific name with family (Leguminosae). Replace “was fed to the predator being reared” with “used to”.
Response: Thanks for this important highlight. We have revised it in our manuscript. Line number 103
12) Line 100: Remove male/female
Response: Thanks for this important suggestion. Done in revised manuscript. Line number 103
13) Line 100: How many Pairs? Replace predators” with “(Male: female)”. Replace “were” with “and”.
Response: Thanks for this imortant highlight. We used 20 Pairs. It has been added in Line number 103.
14) Line 102: Remove laid by female predators. adults
Response: Line number 106
15) Line 103: Remove predators laid egg batches on bean leaves. Batches
Response: Thanks for this important suggestion. Done in revised manuscript. Line number 106
16) Line 105: Eclosing? Remove Predator culture was established by following the above protocol.
Response: Thanks for this technical highlight. We removed Ecolosed to Eclosed as suggested by reviewer 1. Line number 108
17) Line 108: Remove which are originally
Response: Thanks for this important suggestion. Line number 111
18) Line 110: Transparent jar (Material, Company, Reference #, City and state)
Response: Thanks for this critical highlight. We took our experiment material from our university Lab i.e. Key Laboratory of Hubei, Insect Resources Utilization and Sustainable Pest Management, located at Huazhong Agricultural University (HZAU), Wuhan, China.
19) Line 111: Replace “were” with “and”
Response: Thanks for this imporatnt highlight. Revised as suggested by reviewer 1. Line number 113
20) Line 112: Similar sized cylindrical jar (Material, Company, Reference #, City and state)
Response: Thanks for this critical highlight. We took our experiment material from our university Lab i.e. Key Laboratory of Hubei, Insect Resources Utilization and Sustainable Pest Management, located at Huazhong Agricultural University (HZAU), Wuhan, China.
21) Line 115: Similar sized cylindrical jar (Dimension, Company, Reference #, City and state)
Response: Thanks for this critical highlight. We took our experiment material from our university Lab i.e. Key Laboratory of Hubei, Insect Resources Utilization and Sustainable Pest Management, located at Huazhong Agricultural University (HZAU), Wuhan, China.
22) Line 116-119: It needs citation, if this diet is not developed by the authors
Response: Thanks for this technical suggestion. It has been added in revised manuscript. Line number 119
- ul Haq, R.; Khan, J.; Ali, G. Rearing of Spodoptera litura (Fabricius) on different artificial diets and its parasitization with Trichogramma chilonis (Ishii). J. Zool. 2015, 47, 169-175.
23) Line 120: Another box jar (Specifications: materials, dimensions, company Reference #, city, state).
Response: Thanks for this critical highlight. We took our experiment material from our university Lab i.e. Key Laboratory of Hubei, Insect Resources Utilization and Sustainable Pest Management, located at Huazhong Agricultural University (HZAU), Wuhan, China.
24) Line 125: Plastic containers (Specifications: materials, dimensions, company Reference #, city, state).
Response: Thanks for this critical highlight. We took our experiment material from our university Lab i.e. Key Laboratory of Hubei, Insect Resources Utilization and Sustainable Pest Management, located at Huazhong Agricultural University (HZAU), Wuhan, China.
25) Line 125: Sets?? How many,? Set or was a set of 12.
Response: It is total 12 sets. One set contain one plastic container. Line number 128
26) Line 126: Remove before the start of the experiment. Replace predator”…. with Harmonia axyridis”. Replace “adult” with “Adult”
Response: Thanks for this important suggestion. Revised as suggested in manuscript. Line number 129
27) Line 127: Acclimatized or adapted? Replace predator with….H. axyridis. Remove predator
Response: Thanks for this technical suggestion. It is adapted and we revised it in our manuscript. Line number 129-130
28) Line 129: This whole setting…. What setting??
Response: Thanks for mentioning important point. We replaced whole setting with whole apparatus and it means that plastic container and adults of H. axyridis and eggs of S. litura. Line number 132
29) Line 131: Replace “acclimatization” with “adaptation”
Response: Thanks for this valuable suggestion. We replaced it in revised manuscript. Line number 134
30) Line 135: Checked every 12 h interval… Why the egs were not collected every 12h to assure the homogeneity in eclosion? Different instars. How many per instar?
Response: Thanks. The eggs were checked and collected every 12 h. It has been revised in manuscript. Line number 138
31) Line 136-137: (0-6 hours) This protocol needs more explanation. How instars 0-6 h old were obtained ? It is not seem very accurate the that is described.
Response: Thanks for highlighting this important point. We used 0-6 h old instars and for that purpose the old instar were kept individually in a petri dish (6 cm). As they conveted or molted into new stage, they were removed and used for experiment. As the eggs hatched, eclosed larvae were kept individually and reared seprately till desired stage. Line number 139
32) Line 137: Replace the “predators” with “they” and was with “were”
Response: Thanks for this valabe suggestion. We revised it in our manuscript. Line number 140
33) Line 138: Individually ??? what the size of petri dish?
Response: Thanks for this technical suggestiion. We placed each instar/stage individually in a Petri dish (6 cm). This line has been added in revised manuscript also. Line number 141
34) Line 142: Prevent cotton. What was the purpose of this leaf disk?
Response: Thanks for highlighting this important point.This leaf disk was used as a substrate and eggs were placed on the that disk during experiment. We added this line in revised manuscript. Line number 144
35) Line 144: Eggs. What were the eggs conditions? They were viable or were frozen? Please explain.
Response: Thanks for higlighting this important point.The freshly laid eggs were used in the experiment for each density and for each replication. And it has been added in Manuscript also Line number 147
36) Line 144-145: Its means that you exposed to each H. axyridis first instar larval 3,6,10 etc eggs for 24 h and same for other instar and stages?
Response: Yes, we did the same. We exposed them for 24 h.
37) Line 150: Explain what replication consisted of: A total of 240 insects? 1 per instar/satge per prey rate and 48 per temperature?
Response: Thanks for mentionenig this point. A replication consisted of 1 stage against 1 density. As eight different densities (3,6,10,15,20,25,30,35) were given to 1st instar and they were relicated ten times (80 times) for first instar at 1 temperature. Line number 154
38) Line 151: What was the duration of the experiment ? 24 hours? Or until the H. axyridis stage died? Or until the predator consumed all eggs?
Response: Yes, the duration of the experiment was 24 h. The data was recorded after 24 h and predators were allowed to consume prey continously for 24 h.
39) Line 152: then put them back?
Response: No. The predators were removed from the Petri dishes and the consumed or damaged eggs were counted after 24 h.
40) Line 165-166: N0 should be italic.
Response: Thanks for this critical highlight. Done in revised manuscript. Line number 169
41) Line 168-173: Introduction or discussion.
Response: Thanks for this teachnical highlight. We kept this line with deatil at the same plaace as suggested by reviewer 1. Line number 177-184
42) Line 174: Introduction or discussion.
Response: We removed that line from our manuscript as suggested by reviewer.
43) Table 1: What is the reason for this column (Mean)?
Response: This column is describing the effects of different temperatures on Overall mean consumption rate of each stage at different temperatures.
44) Line 224-227: Shold be add in Material and methods and remove predator in line 225.
Response: We removed these line and added in Material and Methods with detail as suggested by reviewer 1. Line number 177-184
45) Line 281-283: Write authority, order and family.
Response: Thanks for this important suggestion. Done in revised manuscript according to valuable suggestion of reviewer. Line number 302-305
References
- Holling, C.S. Some characteristics of simple types of predation and parasitism. Can. Entomol. 1959, 91, 385-398.
- Juliano, S.A. Nonlinear curve fitting: predation and functional response curves; Chapman and hall: New York, 2001; Vol. 2, pp. 178-196.
- Rogers, D. Random search and insect population models. J. Anim. Ecol. 1972, 41, 369-383.
- Koch, R.L.; Hutchison, W.D.; Venette, R.; Heimpel, G.E. Susceptibility of immature monarch butterfly, Danaus plexippus (Lepidoptera: Nymphalidae: Danainae), to predation by Harmonia axyridis (Coleoptera: Coccinellidae). Biol. Control 2003, 28, 265-270.
- Xue, Y.; Bahlai, C.A.; Frewin, A.; Sears, M.; Schaafsma, A.; Hallett, R.H. Predation by Coccinella septempunctata and Harmonia axyridis (Coleoptera: Coccinellidae) on Aphis glycines (Homoptera: Aphididae). Environ. Entomol. 2009, 38, 708-714.
Reviewer 2 Report
The current study investigates the potential use of the ladybird beetle Harmonia axyridis as egg predator of the tobacco cutworm, Spodoptera litura, by evaluating functional responses of the beetle. Egg consumptions of all larval and adult stages of H. axyridis were measured by varying the number of eggs offered and temperature in petri dish arenas and different coefficients (i.e., instantaneous attack rate, handling time, theoretical maximum predation rate) were estimated based on models. Because 4th instar larvae, males and females had higher consumption rate than younger stages, the authors recommend the later stages to control S. litura infestations. While I understand the need to test potential predators for pest and estimate their impact, this study lacks interconnecting the need to estimate these parameters for a pest management program. After reading this manuscript, I imagine that collecting this data and estimating these coefficients would be interesting from a mechanistic or ecological perspective, but I wonder what is the use of these functional responses for a biological control or pest management program. Also, there is no comparison of the authors’ findings to other studies (if any), making it difficult to evaluate the performance of H. axyridis to other species. I recommend the current manuscript to a major revision.
Major comments
Line 175: The authors never explained what the Roger’s random attack model is and why it was used here. Why is this appropriate for this data?
Line 180: the term “searching time” is misleading. The experiment lasted 24 hr and part of this time was spent handling a prey, so the actual time spent searching for a prey has to be less than 24 hr (24 hr – handling time). I suggest changing this term to “duration of experiment”.
Data analysis section: why estimating all these parameters (i.e., instantaneous attack rate, handling time, total searching time) is important for a biocontrol program? How are these parameters in the decision-making of an IPM strategy?
Since the data is not normally distributed, have the authors considered using a Poisson glm? This distribution would be appropriate for counts (i.e., number of eggs consumed). I recommend trying this analysis because it seems odd that a mean prey consumption of, for example, females (65.78 +/- 17.21) is not different than of 1st instar (9.78 +/- 1.31), according to Table 1.
Table 1:
- As I stated before, could the authors re-check the comparisons between all stages at 35 oC? It is strange to me that prey consumptions by all stages is statistically not significant at this temperature (all stages are followed by “A”) but smaller differences at, for example 15 or 20 oC, are different between stages. As suggested before, please check the Poisson distribution for this data.
- The comparisons inside each life stages of 3rd, 4th, male and female at the different temperatures have no mean followed by lower case “a”. This should be the first letter to be used but instead the first letter used is “b”. The same applies for table 2.
Lines 195-196: This was confusing for me. I understand the term “within columns” as the means (+/-SE) inside each temperature but instead lower case letters are used to compare means inside each row or stages (see the row Mean and lower case letters). The same applies for “within rows” (see column Mean and upper case letters). This is confusing and I suggest to be more specific, like this (same for table 2, lines 199 and 200):
Means within the same stage at different temperatures followed by the same lower case letters are not significantly different. Means within the same temperature of different stages followed by the same upper case letters are not significantly different.
Table 2: Comparisons of stages at 20, 25, 30 and 35 oC do not start from letter “A” but from “B”. Correct this.
Lines 208 and 209: On what basis are the authors concluding this? If the statistics are correct and according to table 1, males and females are not statistically different than 3rd instar. By table 2, 4th instar, males and females are not statistically different than 2nd instar, and 4th instar is not different than 3rd instar. So, why 4th instar larvae, males and females are considered equally superior than other stages?
Table 3: why not forcing all intercepts to be 0? If there is no prey, then there is no consumption. Why allowing a model to estimate a positive or negative consumption at no prey? Biologically speaking, it doesn’t make sense.
Line 226: Like I mentioned before, what is Roger’s random predator equation? Reference?
Line 231: Could you define what attack rate is? Is it number of eggs consumed/h? Attacks/h (what is the definition of attack here? When does the attack start and end? Attack to a batch of eggs? Sounds odd to me that a female would launch a 0.546 attack/h (at 35 oC). In this case, for instance, what is 0.546 attack?
On the same topic, how accurate are these coefficients compared to observed attack rates? Is there any reference for this?
Line 261: I don’t understand the importance of the theoretical maximum predation rate. T is fixed at 24 hr, so TMP is just an inverse measure of handling time, i.e. if Th is small, then TPM will be large. So, why is this parameter important if handling time is known?
Line 326: The authors mention that H. axyridis has a high attack rate but there is no comparison to other predators or a threshold to suggest that this is a high rate. Could the authors compare their findings to other studies and define a “high rate”?
Line 344, last sentence: It sounds to me that 4th instar and adults is being recommended for pest management based on their consumption rate, dismissing the potential of other larval stages. I have two issues with this. First, how can all the other estimated parameters obtained here assist the decision-making in pest management? If the conclusion is going to be based solely on consumption rate, what is the relevance of studying all the parameters presented in this study? Second, 4th instar larvae will soon pupate and won’t be active and adults will spend part of their time searching for mates and ovipositing, so these stages have some disadvantages being inactive for a while. I think the conclusion section should be better developed to consider different scenarios of pest management. For instance, can younger stages be released at lower pest infestations, while older stages be used for higher infestations?
Minor comments:
Line 59: “with over 60 share” is not clear. Replace by “corresponding to over 60%”.
Line 60: change “environmental” to “the environment”
Line 62: chemical insensitivity is one type of resistance. There are other types of insecticide resistance. Check the IRAC webpage: https://irac-online.org/about/resistance/mechanisms/. Delete “due to loss of chemical insensitivity” or change accordingly.
Line 72: delete “as”
Line 87: Add “The” before “current”, and replace “was initiated with aim” by “aimed”.
Line 93: H. axyridis in italic.
Line 105: Eclosed, not ecolosed
Line 107: S. litura in italic
Line 109: Add “The” before “Institute”
Line 110: Move “in a circular transparent jar (21cm height x 9 cm diameter)” to after “kept” (line 111).
Line 137: Do you mean you placed one individual per dish? Authors need to clarify in the text.
Line 151: “damaged”, not “damage”
Line 190: Add “(Table 1)” to after “35 oC”.
Line 224: Add “(fig. 1)” after “density”.
Line 228: Should this sentence refer to table 4, instead of tables 1 and 2?
Line 230: Add “(table 3)” after “parameters”.
Line 275: Add a reference to after “crop pests”.
Line 287: Is it “pest biology” or “predator biology”?
Line 288: Add “and are” before “likely”.
Line 302: Delete “by”.
Line 311: Delete “of”.
Line 328: Change “However” for “Although”.
References: There are many mistakes in the reference list. Here are a few that I spotted but I suggest the authors to fully revise their list as there may be more.
Line 477: The names Harmonia axyiridis and Rhopalosiphum prunifoliae should be italicized.
Lines 487, 509, 538, 545, 555: The journal names should not be all in capital letters.
Line 510: variegata should be lower case.
Line 511: septempunctata should be lower case. “to the aphid” should not be in italic. Schizaphis graminum should be in italic. Graminum in lower case.
Line 518: Harmonia axyridis and Rhopalosiphum numphaeae in italic.
Line 525: Podisus in italic
Line 532: Journal name is incomplete. Oecologia.
Line 540: Space missing between (Pallas) and (Coleoptera…)
Line 547: I have not checked the author guidelines for references but usually book titles are preceded by “In:”
Line 549: Hyphen missing between “density” and “dependent”
Line 549: Journal name incomplete. Phytoparasitica.
Author Response
Dear Editor and Reviewer,
On behalf of all authors, I am thankful for valuable comments and suggestions on the manuscript insects-865851, entitled "Temperature dependent functional response of Harmonia axyridis (Coleoptera: Coccinellidae) on the eggs of Spodoptera litura (Lepidoptera: Noctuidae) in laboratory". These comments are valuable and helpful for revising and improving the manuscript, as well as this is an important guidance for our future research.
Reply to Comments of the editor
- Improvement to writing and data presentation
Reply: Thanks for your valuable suggestions and technical comments. These comments really helped us and improved the quality of our manuscript. We thoroughly re-checked all of our analysis, references and improved them according to your suggestions.
- Assess the effect of temperature on the prey as well, in addition to the beahviour of the predator
Reply: The predaror and prey stock cultures were maintained at same condition (27 ± 2 °C and 70 ± 5 % RH with a photoperiod of 14:10 h (L:D) throughout the experiment. The eggs and one predator per petri plate were exposed to the desired remperature for the duration of the experiment (24 h). Since the actual evaluation time was short, therefore no appreciable effect is expected on the growth of the prey. Since we used eggs as prey, no change is expected in their behavior during the short period of the experinment.
And the responses to Reviewer 2's comments have been attached, check it please. Thank you!
-------------------------------------------
Reviewer 2
General comments:
- While I understand the need to test potential predators for pest and estimate their impact, this study lacks interconnecting the need to estimate these parameters for a pest management program. After reading this manuscript, I imagine that collecting this data and estimating these coefficients would be interesting from a mechanistic or ecological perspective, but I wonder what is the use of these functional responses for a biological control or pest management program.
Response: Thanks for mentioning this important point. Functional response is an imperative criterion for accessing the efficency of a predator on a given prey. Handling time and attack rate also called (search rate or searching time or rate of discovery or space clearance rate) areimportant parametersof the functional response. The attack rate determines the ability of a predator to catch prey in a given time and handling time indicates the time a predator spends to identify, subjugate, attack and consume a particular prey [1]. These parameters determines the foraging behavior of a predator and tell how much efficient is a predator against a given density of prey. As the foraging behavior increase the attack rate will also increase which leads to decrease in handling time. It means the predator will consume more prey and we can use it to control pest. The parameters of functional response help to explain relative changes in the efficacy of the predator with changing density of the prey, along the growth of the predator. For example, early in the season when the pest density is low, the predator efficincy is also axpected to be low, based on the parametes of functional response. Similarly, as the predator grows through different instars, the functional reponse parameters aslo improve. Therefore, the capacity of the predator to regulate the pest population will improve. Therefore, the functinal repsonse parametres give an indication of the temporal variation in the efficcacy of the predator to rgulate the pest population.
- Holling, C.S. Some characteristics of simple types of predation and parasitism. Entomol. 1959, 91, 385-398.
- Also, there is no comparison of the authors’ findings to other studies (if any), making it difficult to evaluate the performance of H. axyridis to other species. I recommend the current manuscript to a major revision.
Response: Thanks for this valuable recommendation.
Appropriate literature has been added in the MS (line numbers 316-319, line numbers 324-329 ) and comparisons made.
Response: In our study we observed highest attack rate for female i.e. 0.546 ± 0.058 h-1. Similarly, Koch and Xue observed high attack rate in female i.e. 0.94 ± 0.374 h-1 and 0.78 h-1 when they allowed H. axyridis to feed on D. plexippus eggs and A. glycines respectively.
Likewise, fourth instar and female of Harmonia variegata showed highest attack rate against A. craccivora i.e. 0.144 ± 0.013 h-1 and 0.106 ± 0.011 h-1 respectively. We noticed lower handling time for fourth instar followed by adult female and male i.e. 0.174 ± 0.003 h, 0.189 ± 0.004 h and 0.243 ± 0.05 h respectively, and Seko reported lower handling time for adults and fourth instar also, when he fed H. axyridis on M. persicae. He found lower handling time for female (0.127 h), followed by male and fourth instar i.e. (0.146 h) and (0.156 h) respectively. In the same way, another researcher reported lower handling time for H. variegata female followed by fourth instar and male i.e. (0.409 ± 0.048 h), (0.454 ± 0.028 h) and (1.194 ± 0.069 h) respectively.
Major Comments:
1) Line 175: The authors never explained what the Roger’s random attack model is and why it was used here. Why is this appropriate for this data?
Response: The Roger’s random predator equation is the integrated form of the disk equation. This equation is appropriate for modelling predation or parasitism whenever predation or parasitism result in a significant reduction of prey or host densities. We did not replace prey in our experiment, that’s why we used Roger’srandom attack model instead of disk equation, where we replace consumed/damaged prey with new prey. Also, we got linear estimates significantly negative for all stages at all tested temperatures, which shows type II functional response and the random predator equation is more appropriate for experimental data of type II functional response [2] for estimation of the parameters of functional response.
The appropriate information and rationale for using this model for parameter estimation has been added in the MS, line numbers 176-182.
- Juliano, S.A. Nonlinear curve fitting: predation and functional response curves; Chapman and hall: New York, 2001; Vol. 2, pp. 178-196.
2) Line 180: the term “searching time” is misleading. The experiment lasted 24 hr and part of this time was spent handling a prey, so the actual time spent searching for a prey has to be less than 24 hr (24 hr – handling time). I suggest changing this term to “duration of experiment”.
Response: Thanks for your technical suggestion. It has been changed in revised manuscript as “T is the duration of experiment (24 h)”. line number 188.
3) Data Analysis Section: why estimating all these parameters (i.e., instantaneous attack rate, handling time, total searching time) is important for a biocontrol program? How are these parameters in the decision-making of an IPM strategy?
Response: Thanks for mentioning this important point. Functional response is an imperative criterion for accessing the efficency of a predator on a given prey. Handling time and attack rate also called (search rate or searching time or rate of discovery or space clearance rate) are important parametersof the functional response. The attack rate determines the ability of a predator to catch prey in a given time and handling time indicates the time a predator spends to identify, subjugate, attack and consume a particular prey [1]. These parameters determines the foraging behavior of a predator and tell how much efficient is a predator against a given density of prey. As the foraging behavior increase the attack rate will also increase which leads to decrease in handling time. It means the predator will consume more prey and we can use it to control pest. The parameters of functional response help to explain relative changes in the efficacy of the predator with changing density of the prey, along the growth of the predator. For example, early in the season when the pest density is low, the predator efficincy is also axpected to be low, based on the parametes of functional response. Similarly, as the predator grows through different instars, the functional reponse parameters aslo improve. Therefore, the capacity of the predator to regulate the pest population will improve. Therefore, the functinal repsonse parametres give an indication of the temporal variation in the efficcacy of the predator to rgulate the pest population.
- Holling, C.S. Some characteristics of simple types of predation and parasitism. Entomol. 1959, 91, 385-398.
4) Since the data is not normally distributed, have the authors considered using a Poisson glm? This distribution would be appropriate for counts (i.e., number of eggs consumed). I recommend trying this analysis because it seems odd that a mean prey consumption of, for example, females (65.78 +/- 17.21) is not different than of 1st instar (9.78 +/- 1.31), according to Table 1.
Response: Table 1 represents the results of the effect of tempearture and predator stage on the average prey consumption. We evluated the effect by non-parametric ANOVA given the data are not normal.
As suggested, we tried generalised linear model with poisson distribution and found that the data are over-dispersed (Sample variance >> Mean). Therefore, we fitted the negative binomial glm to the data followed by ANOVA and Tukey HSD. The results differ considerably from the earlier analysis. The results of the new analysis are given in Table 1.
5) Table 1: As I stated before, could the authors re-check the comparisons between all stages at 35oC? It is strange to me that prey consumptions by all stages is statistically not significant at this temperature (all stages are followed by “A”) but smaller differences at, for example 15 or 20 oC, are different between stages. As suggested before, please check the Poisson distribution for this data.
Response: The data on average prey consumption were analysed with negative binomial glm followed by ANOVA and tukey HSD. The significance codes have aslo been corrected in table 1.
6) Table 2: The comparisons inside each life stages of 3rd, 4th, male and female at the different temperatures have no mean followed by lower case “a”. This should be the first letter to be used but instead the first letter used is “b”. The same applies for table 2.
Response: The lettering has been corrected in table 2. The significance codes begin with “a” on the smallest value and so on.
7) Lines 195−196: This was confusing for me. I understand the term “within columns” as the means (+/-SE) inside each temperature but instead lower case letters are used to compare means inside each row or stages (see the row Mean and lower case letters). The same applies for “within rows” (see column Mean and upper case letters). This is confusing and I suggest to be more specific, like this (same for table 2, lines 199 and 200).
Response: The lettering has been corrected in table 1 and 2. Means within the same column followed by the same letter (lower case) are not significantly different (Tukey’s HSD Test, p < 0.05). Means within the same row followed by the same letter (upper case) are not significantly different (Tukey’s HSD Test, p < 0.05). the asterisk (*) is used to denote that the effect of temperature non-significant on the average prey consumption of certain instars (Tukey’s HSD Test, p > 0.05).
Same legend is used for the prportionate prey consumtion based on Kruskal-Wallis Test, and Dunn’s Multiple Comparison Test, p = 0.05).
8) Table 2: Comparisons of stages at 20, 25, 30 and 35 oC do not start from letter “A” but from “B”. Correct this.
Response: The significance codes have been corrected. The significance codes begin with “a” on the smallest value and so on.
9) Lines 208 and 209:On what basis are the authors concluding this? If the statistics are correct and according to table 1, males and females are not statistically different than 3rd instar. By table 2, 4th instar, males and females are not statistically different than 2nd instar, and 4th instar is not different than 3rd instar. So, why 4th instar larvae, males and females are considered equally superior than other stages?
Response: We revised our tables and results and corrected according to table. line number 213-214.
10)Table 3: why not forcing all intercepts to be 0? If there is no prey, then there is no consumption. Why allowing a model to estimate a positive or negative consumption at no prey? Biologically speaking, it doesn’t make sense.
Response: The polynomial logistic regression is used to identify thye type of functional response (type II or type III) (Juliano, 2001). The signs of linear and quadratic coefficients indicate the type of reposne. This information quantifies the type of change in the proportionate prey consumtion against offered density of prey, and not the positive or negative comsumption. Therefore, we need not interfere with the intercepts.
- Juliano, S.A. Nonlinear curve fitting: predation and functional response curves; Chapman and hall: New York, 2001; Vol. 2, pp. 178-196.
11) Line 226: Like I mentioned before, what is Roger’s random predator equation? Reference?
Response: The Roger’s random predator equation [3] is the integrated form of the disk equation. This equation is appropriate for modelling predation or parasitism whenever predation or parasitism result in a significant reduction of prey or host densities. The equation is:
where Ne=number of prey eaten, No=number of prey present at the start of the experiment, T=total time of the experiment (24 h in this experiment), Th = handling time per prey item, and a = the attack constant, or instantaneous rate of discovery.
We removed that line from our results as suggested by reviewer 4, and we have already described that line in material and methods. line numbers 176-182.
- Rogers, D. Random search and insect population models. Anim. Ecol. 1972, 41, 369-383.
12) Could you define what attack rate is? Is it number of eggs consumed/h? Attacks/h (what is the definition of attack here? When does the attack start and end? Attack to a batch of eggs? Sounds odd to me that a female would launch a 0.546 attack/h (at 35 oC). In this case, for instance, what is 0.546 attack?
On the same topic, how accurate are these coefficients compared to observed attack rates? Is there any reference for this?
Response:The attack rate determines the ability of a predator to catch prey in a given time [1]. line numbers 315.
It starts when a predator catch a prey for killing. Here, the attack rate means the attack of a predator on the eggs of prey.
- Holling, C.S. Some characteristics of simple types of predation and parasitism. Entomol. 1959, 91, 385-398.
13) Line 261: I don’t understand the importance of the theoretical maximum predation rate. T is fixed at 24 hr, so TMP is just an inverse measure of handling time, i.e. if Th is small, then TPM will be large. So, why is this parameter important if handling time is known?
Response: Actually, the maximum predation rate (T/Th) represents the maximum number of prey that can be consumed by an individual during a given time (24 hr). Yes, there is a inverse relation between handling time and maximum predation rate. From this parameter we can find how many prey could be consumed by a predator. It is derived as a fitted parameter from the estimates of handling time ftom the Rogers’ reamdom predator equation (Rogers, 1972).
14) Line 326: The authors mention that H. axyridis has a high attack rate but there is no comparison to other predators or a threshold to suggest that this is a high rate. Could the authors compare their findings to other studies and define a “high rate”?
Response: Thanks for your technical recommendation. The attack rate is the parameter of functional response and it vary from predator to predator, predator stage, arena size, temperature and prey size also. It is not fixed for any predator or predator stage. In our study we observed highest attack rate for female i.e. 0.546 ± 0.058 h-1. Similarly, Koch [4] observed high attack rate in female .i.e. 0.94 ± 0.374h-1, when he allowed Harmonia axyridis to feed on Danaus plexippus eggs. Likewise, 0.78 h-1 attack rate was noted by Harmonia axyridis on Aphis glycines[5].
The necessary information on comparison anjd the appropriate references have been added to the Manuscript at line numbers 326-331.
- Koch, R.L.; Hutchison, W.D.; Venette, R.; Heimpel, G.E. Susceptibility of immature monarch butterfly, Danaus plexippus (Lepidoptera: Nymphalidae: Danainae), to predation by Harmonia axyridis (Coleoptera: Coccinellidae). Control 2003, 28, 265-270.
- Xue, Y.; Bahlai, C.A.; Frewin, A.; Sears, M.; Schaafsma, A.; Hallett, R.H. Predation by Coccinella septempunctata and Harmonia axyridis (Coleoptera: Coccinellidae) on Aphis glycines (Homoptera: Aphididae). Entomol. 2009, 38, 708-714.
15) Line 344, last sentence: It sounds to me that 4th instar and adults is being recommended for pest management based on their consumption rate, dismissing the potential of other larval stages. I have two issues with this. First, how can all the other estimated parameters obtained here assist the decision-making in pest management? If the conclusion is going to be based solely on consumption rate, what is the relevance of studying all the parameters presented in this study? Second, 4th instar larvae will soon pupate and won’t be active and adults will spend part of their time searching for mates and ovipositing, so these stages have some disadvantages being inactive for a while. I think the conclusion section should be better developed to consider different scenarios of pest management. For instance, can younger stages be released at lower pest infestations, while older stages be used for higher infestations?
Response: As you clearly pointed out, the higher efficacy of 4th instars and adults is not inferred to be recommnded for IPM in the MS, instead, they are deemed as most efficient among all the stages. It measn that these stages will regulate the pest population in a ‘time’ and ‘per unit’ efficient way. Therefore, we would expect much of the pest control will be provided by the 4th instars and adults. As you pointed out, it would be advisable to release early instar predators coincing with the egg laying period (i.e. early infestation) of tobacco cutworm. As has been pointed out the conclusion section of the MS, studies on functional response are not sufficient to develop a filed recommendation however, theese kinds of studies assist in selection appropriate predator species and stages intended for biocontrol. Also, they indicate which stage of the target pest would be best controlled with a specific species and stage of the predator.
Further we found that the predators can do well at higher tempertatures (up to 35° C) which indicates that the efficiency of the predators is going to improve with advance in crop season and can be utilized in protected cultivation where the temperature is typically higher than the ambient.
Minor Comments:
1) Line 59: “with over 60 share” is not clear. Replace by “corresponding to over 60%”.
Response: Thanks for this important suggestion. Corrected as suggested, in revised manuscript. Line number 62
2) Line 60: change “environmental” to “the environment”
Response: Thanks for highlighting this point. It has been corrected in revised manuscript. Line number 63
3) Line 62: chemical insensitivity is one type of resistance. There are other types of insecticide resistance.
4) Check the IRAC webpage: http://iraconline.org/about/resistance/mechanisms/. Delete “due to loss of chemical insensitivity” or change accordingly.
Response: Thank you for mentioning this point. It has been replaced with “due to development of insecticide”. Line number 65
5) Line 72: delete “as”
Response: Done in revised manuscript according to suggestion of reviewer. Line number 75
6) Line 87: Add “The” before “current”, and replace “was initiated with aim” by “aimed”.
Response: Thanks for highlighting this important point.Corrected in revised manuscript. Line number 90
7) Line 93: H. axyridis in italic.
Response: Thanks for this critical highlight. Corrected in revised manuscript. Line number 96
8) Line 105: Eclosed, not ecolosed
Response: Thanks for this techanical suggestion. It has been corrected in revised manuscript. Line number 107
9) Line 107: S. litura in italic
Response: Thanks for this critical highlight. Corrected in revised manuscript. Line number 109
10) Line 109: Add “The” before “Institute”
Response: Thanks for this valuable recommendation. It has been added in revised manuscript. Line number 110
11) Line 110: Move “in a circular transparent jar (21cm height x 9 cm diameter)” to after “kept” (line 111).
Response: Thanks for this important recommendation. Done in revised manuscript. Line number 112
12) Line 137: Do you mean you placed one individual per dish? Authors need to clarify in the text.
Response: Thanks for this important technical point. Yes we kept them individually and we added in revised manuscript. Line number 139
13) Line 151: “damaged”, not “damage”
Response: Thanks for this important highlight. Done in revised manuscript. Line number 153
14) Line 190: Add “(Table 1)” to after “35 oC”.
Response: Thanks for this important suggestion. Done in revised manuscript. Line number 199
15) Line 224: Add “(fig. 1)” after “density”.
Response: Thanks for this technical suggestion. Corrected in revised manuscript. Line number 230
16) Line 228: Should this sentence refer to table 4, instead of tables 1 and 2?
Response: We revised our results and tables and removed that line from manuscript.
17) Line 230: Add “(table 3)” after “parameters”.
Response: Thanks for this technical suggestion. Done in revised manuscript. We placed our table after parameters but we replaced some lines as suggested by reviewer 4.
18) Line 275: Add a reference to after “crop pests”.
Response: Thanks for this valueable suggestion. Added in revised manuscript. Line number 293
19) Line 287: Is it “pest biology” or “predator biology”?
Response: Thanks for this important highlight. It is predator biology and has been corrected in revised manuscript. Line number 307
20) Line 288: Add “and are” before “likely”.
Response: Thanks for this techanical suggestion. It has been corrected in revised manuscript. Line number 308
21) Line 302: Delete “by”.
Response: Done in revised manuscript according to valuable suggestion of reviewer. Line number 331
22) Line 311: Delete “of”.
Response: This word has been removed in revised manuscript. Line number 340
23) Line 328: Change “However” for “Although”.
Response: Thanks for this suggestion. Corrected in revised manuscript. Line number 362
References
24) There are many mistakes in the reference list. Here are a few that I spotted but I suggest the authors to fully revise their list as there may be more.
Response: Thanks for this critical highlight. It has been corrected in revised manuscript.
25) Line 477: The names Harmonia axyiridis and Rhopalosiphum prunifoliae should be italicized.
Response: Thanks for this critical highlight. Corrected in revised manuscrpt. Line number 502
26) Lines 487, 509, 538, 545, 555: The journal names should not be all in capital letters.
Response: We are thankful to you for highlighting this point. Corrected in revised manuscrpt.
Now they are in line numbers 524, 541, 567, 581 and 591 respectively.
27) Line 510: variegata should be lower case.
Response: Corrected in revised manuscrpt. Line number 542
28) Line 511: septempunctata should be lower case. “to the aphid” should not be in italic. Schizaphis graminum should be in italic. Graminum in lower case.
Response: Corrected in revised manuscrpt. Line numbers 542-543
29) Line 518: Harmonia axyridis and Rhopalosiphum numphaeae in italic.
Response: Corrected in revised manuscrpt. Line number 548
30) Line 525: Podisus in italic
Response: Thanks for this important technical point. Corrected in revised manuscrpt. Line number 555
31) Line 532: Journal name is incomplete. Oecologia.
Response: Thanks for mentioning this point. Corrected in revised manuscrpt. Line number 561
32) Line 540: Space missing between (Pallas) and (Coleoptera…)
Response: Thanks for mentioning this point. Corrected in revised manuscrpt. Line number 570
33) Line 547: I have not checked the author guidelines for references but usually book titles are preceded by “In:”
Response: Thanks for this critical highlight. Corrected in revised manuscrpt. Line number 583
34) Line 549: Hyphen missing between “density” and “dependent”
Response: Thanks for this critical highlight. Corrected in revised manuscrpt. Line number 585
35) Line 549: Journal name incomplete. Phytoparasitica.
Response: Thanks for this important technical point. Corrected in revised manuscrpt. Line number 586
Reviewer 3 Report
Dear Authors,
I think that you carried out a nice and interesting study. In the text there are some minor details that you should correct. However, I have several concerns related with the references. Actually, I have detected plenty of typos and mistakes in the cited references that generate certain reservations and, unnecessarily detract your work. All of them must be corrected prior approval for publication.
I terms of the usefulness of your results, I have a big concern that I would like to share with you:
I understand that you are interested in knowing the response of the predator. However, I am wondering what happens to the prey in relation to the increase in temperature? This is not included in the model. However, this must be taken into account and extrapolate under experimental conditions. I am quite sure that the prey also develops faster as the temperature increases. If so, I am not convinced that: “We can use 4th instar and adults in green houses and fields to control this notorious pest.” 

Finally, it is important that you consider that normality must be validated on the errors of the proposed linear model rather than on the data.

Author Response
Dear Editor and Reviewer,
On behalf of all authors, I am thankful for valuable comments and suggestions on the manuscript insects-865851, entitled "Temperature dependent functional response of Harmonia axyridis (Coleoptera: Coccinellidae) on the eggs of Spodoptera litura (Lepidoptera: Noctuidae) in laboratory". These comments are valuable and helpful for revising and improving the manuscript, as well as this is an important guidance for our future research.
Reply to Comments of the editor
- Improvement to writing and data presentation
Reply: Thanks for your valuable suggestions and technical comments. These comments really helped us and improved the quality of our manuscript. We thoroughly re-checked all of our analysis, references and improved them according to your suggestions.
- Assess the effect of temperature on the prey as well, in adiition to the beahviour of the predator
Reply: The predaror and prey stock cultures were maintained at same condition (27 ± 2 °C and 70 ± 5 % RH with a photoperiod of 14:10 h (L:D) throughout the experiment. The eggs and one predator per petri plate were exposed to the desired remperature for the duration of the experiment (24 h). Since the actual evaluation time was short, therefore no appreciable effect is expected on the growth of the prey. Since we used eggs as prey, no change is expected in their behavior during the short period of the experinment.
And the responses to Reviewer 3's comments have been attached, check it please. Thank you!
------------------------------------------
Reviewer 3
General Comments: I think that you carried out a nice and interesting study. In the text there are some minor details that you should correct. However, I have several concerns related with the references. Actually, I have detected plenty of typos and mistakes in the cited references that generate certain reservations and, unnecessarily detract your work. All of them must be corrected prior approval for publication.
I terms of the usefulness of your results, I have a big concern that I would like to share with you:
Response: Thanks for your kind suggestions and important highlights. We followed these suggestions and answered scientifically also. These suggestions are really helpful for us and has improved our manuscripit.
1) I have detected plenty of typos and mistakes in the cited references that generate certain reservations and, unnecessarily detract your work. All of them must be corrected prior approval for publication.
Response: Thanks for yourtechnicalhighlights, we have revised all references according to your suggestions and journal format and it has improved our revised manuscrpit.
2) I understand that you are interested in knowing the response of the predator. However, I am wondering what happens to the prey in relation to the increase in temperature? This is not included in the model. However, this must be taken into account and extrapolate under experimental conditions. I am quite sure that the prey also develops faster as the temperature increases. If so, I am not convinced that: “We can use 4th instar and adults in green houses and fields to control this notorious pest.”
Response: Thanks for you valuable suggestions. As, the duration of our experiment was 24 hours and firstly we placed freshly laid eggs at all tested temperatures in preliminary experiment and observed them for more than 2 days and we found that there was no hatching at all temperatures. And secondly, the eggs are static and can not move. Therefore, we did not consider the effect of temperature on prey/eggs.
3) Finally, it is important that you consider that normality must be validated on the errors of the proposed linear model rather than on the data.
Response: The normality of errors was taken care of in both negative binomial glm and non-parametric anova.
References
4) Theare are many mistakes in references.
Response: Thanks for this technical higlight. We have revised our References according to your technical highlights and journal format.
Scientific Names
5) Write full Scientific names of Predator (Harmonia axyridis) and Prey (Spodoptera litura) in title and in legends of Tables.
Response: Thanks for this technical higlight. We have revised our References according to your technical highlights and journal format.
6) Line 123: Don’t italic Functional Response.
Response: Thanks for this technical higlight. Line number 125
Reviewer 4 Report
Dear Editor,
I have read it, I found it, it is good topic dealing with intessing theme, it is accepted as it is
Best regards
Author Response
Dear Editor and Reviewer,
On behalf of all authors, I am thankful for valuable comments and suggestions on the manuscript insects-865851, entitled "Temperature dependent functional response of Harmonia axyridis (Coleoptera: Coccinellidae) on the eggs of Spodoptera litura (Lepidoptera: Noctuidae) in laboratory". These comments are valuable and helpful for revising and improving the manuscript, as well as this is an important guidance for our future research.
Reply to Comments of the editor
- Improvement to writing and data presentation
Reply: Thanks for your valuable suggestions and technical comments. These comments really helped us and improved the quality of our manuscript. We thoroughly re-checked all of our analysis, references and improved them according to your suggestions.
- Assess the effect of temperature on the prey as well, in adiition to the beahviour of the predator
Reply: The predaror and prey stock cultures were maintained at same condition (27 ± 2 °C and 70 ± 5 % RH with a photoperiod of 14:10 h (L:D) throughout the experiment. The eggs and one predator per petri plate were exposed to the desired remperature for the duration of the experiment (24 h). Since the actual evaluation time was short, therefore no appreciable effect is expected on the growth of the prey. Since we used eggs as prey, no change is expected in their behavior during the short period of the experinment.
And the response to Reviewer 4's comments has been attached, check it please. Thank you!
-----------------------------------------
General Comments:
Dear Editor,
I have read it, I found it, it is good topic dealing with intessing theme, it is accepted as it is.
Response: Thank you very Much.
Round 2
Reviewer 1 Report
Please find attached file with minor suggestions.

Author Response
Dear Editor and Reviewer,
On behalf of all authors, I am thankful for valuable comments and suggestions on the manuscript insects-865851, entitled "Temperature dependent functional response of Harmonia axyridis (Coleoptera: Coccinellidae) on the eggs of Spodoptera litura (Lepidoptera: Noctuidae) in laboratory". These comments are valuable and helpful for revising and improving the manuscript, as well as this is an important guidance for our future research.
Reply to the editor
Thanks. We addressed all the questions and changes in revised manuscript. And sorry we could not mention correct line numbers in response letter in previous revision. It happened due to copy paste problem and I think one reviewer review our file in PDF format and other reviewer in word. To overcome this problem, we are attaching two files. One is in word format with track changes and same file with accepted track changes in word format to overcome repetion problem., so that reviewer can see that there is no repetition. Because in PDF file that was sent to us for further revision, two paragraphs were two times but they were not present in Word file.
Secondly, We improved English and removed grammatical mistakes and made some changes in discussion for coherence as suggested by one reviewer with track changes.
If you think, this manuscript needs some changes more then let us know. Thanks Again!
Instructios for Reviewer: Please follow the word file for review with track changes not the PDF version.
------------------------------------------------------------
We replaced order of Acyrthosiphon pisum from Homoptera to Hemiptera. We made this changes in Materials and Methods. LINE NO 96
Now the order for this pest has changed. The reference is attached below:
Ximenez-Embuna, M.G. et al. (2014). Seasonal, spatial and diel partitioning of Acyrthosiphon pisum (Hemiptera: Aphididae) predators and predation in alfalfa fields. Biological Control 69, 1-7.
Reviewer #1
Please follow the word file for review with track changes not PDF.
1) Line 103: “20” with “Twenty”.
Response: Corrected according to the reviewer suggestion. Line Number (102)
2) Line 104: Replace “of 60 cm length, 44 cm width and 34 cm height” with “(60 x 44 x 34 cm)”.
Response: Corrected according to the reviewer suggestion. Line Number (103)
3) Line 106: Add were after female. Add H. axyridis and with Eggs batches”.
Response: Corrected according to the reviewer suggestion. Line Number (103)
4) Line 113, 115 and 123: Cross out cm height and explain the material of jar either it was made up of plastic or glass?
Could you provide the manufacture, reference number, city and state or cite a webpage. Same for the other containers that are mentioned in M&M. This was suggested in the last revision.
Response: Done according to the reviewer suggestions and added some important information in revised manuscript also.
Circular glass transparent jar (21 × 9 cm diameter) Line Number (112-113)
Company Name: Sichuan Shubo (Group) Co., Ltd.
Model No. GG-17
Company Registration Number: 510184000015999
Link: https://m.baike.so.com/doc/12266144-12803597.html
Plastic Box (12 × 7.5 cm diameter) Line Number (123-124)
Company Name: Wenling Daxi Lingping Plastic Products Factory
Company Registration Number: 331081100094564
This is a Local Company and there was no Model No for this.
Link: https://m.tianyancha.com/company/2320994515
5) Line 146: What was the function of Cotton leaf disk?
Response: A single fresh cotton leaf disk was centered upside down on the agar solution to obtain uniform leaf surface. Secondly, the mid rib of leaf was disturbing and not fitting on solidify agar solution. Therefore, we put it upside down. Line Number (148)
6) Line 170: N0 should be italic.
Response: Corrected according to the reviewer suggestion. Line Number (174)
7) Line 177-178: Cross out this line.
Response: We cut these lines and arranged as suggested by reviewer. Line Number (182-184)
8) Line 182-183: Write these lines in results
Response: Thanks for this important highlight. Revised as suggested by reviewer. Line Number (222-224)
9) Line 184-186: Cross out this line.
Response: Done.
10) Line 224: Replace “Harmonia” with “H.”.
Response: Done. Line Number (221)

Reviewer 2 Report
General comments:
The manuscript was greatly improved and I have just a few more comments. Please be aware that some of my comments were not addressed, although the authors said that they were, and I am posting them again. In addition, authors cited wrong line numbers to the great majority of additions and corrections that they made. Some of them were not even close to the right line number. It took me a while to find them all and it is very time consuming. Before resubmitting your revised manuscript and letter, the last thing the authors should do is making sure that all line numbers are correct in their response letter.
Major Comments:
3) Data Analysis Section: why estimating all these parameters (i.e., instantaneous attack rate, handling time, total searching time) is important for a biocontrol program? How are these parameters in the decision-making of an IPM strategy?
Response: Thanks for mentioning this important point. Functional response is an imperative criterion for accessing the efficency of a predator on a given prey. Handling time and attack rate also called (search rate or searching time or rate of discovery or space clearance rate) are important parametersof the functional response. The attack rate determines the ability of a predator to catch prey in a given time and handling time indicates the time a predator spends to identify, subjugate, attack and consume a particular prey [1]. These parameters determines the foraging behavior of a predator and tell how much efficient is a predator against a given density of prey. As the foraging behavior increase the attack rate will also increase which leads to decrease in handling time. It means the predator will consume more prey and we can use it to control pest. The parameters of functional response help to explain relative changes in the efficacy of the predator with changing density of the prey, along the growth of the predator. For example, early in the season when the pest density is low, the predator efficincy is also axpected to be low, based on the parametes of functional response. Similarly, as the predator grows through different instars, the functional reponse parameters aslo improve. Therefore, the capacity of the predator to regulate the pest population will improve. Therefore, the functinal repsonse parametres give an indication of the temporal variation in the efficcacy of the predator to rgulate the pest population.
- Holling, C.S. Some characteristics of simple types of predation and parasitism. Entomol. 1959, 91, 385-398.
Reviewer: this is a good response. I think the manuscript would gain if you adapted this response into the Discussion section. Please consider this request. Also, be aware that there are several typos in your response, in case you decide to incorporate it into your manuscript.
4) Since the data is not normally distributed, have the authors considered using a Poisson glm? This distribution would be appropriate for counts (i.e., number of eggs consumed). I recommend trying this analysis because it seems odd that a mean prey consumption of, for example, females (65.78 +/- 17.21) is not different than of 1st instar (9.78 +/- 1.31), according to Table 1.
Response: Table 1 represents the results of the effect of tempearture and predator stage on the average prey consumption. We evluated the effect by non-parametric ANOVA given the data are not normal.
As suggested, we tried generalised linear model with poisson distribution and found that the data are over-dispersed (Sample variance >> Mean). Therefore, we fitted the negative binomial glm to the data followed by ANOVA and Tukey HSD. The results differ considerably from the earlier analysis. The results of the new analysis are given in Table 1.
Reviewer: Thank you for the pro-active action of analyzing the data using negative binomial once the poisson was not appropriate! Well done!
12) Could you define what attack rate is? Is it number of eggs consumed/h? Attacks/h (what is the definition of attack here? When does the attack start and end? Attack to a batch of eggs? Sounds odd to me that a female would launch a 0.546 attack/h (at 35 oC). In this case, for instance, what is 0.546 attack?
On the same topic, how accurate are these coefficients compared to observed attack rates? Is there any reference for this?
Response:The attack rate determines the ability of a predator to catch prey in a given time [1]. line numbers 315.
It starts when a predator catch a prey for killing. Here, the attack rate means the attack of a predator on the eggs of prey.
- Holling, C.S. Some characteristics of simple types of predation and parasitism. Entomol. 1959, 91, 385-398.
Reviewer: A response to my second question about the accuracy of these coefficients compared to actual attack rates is missing. Any thoughts?
Minor Comments:
Reviewer: About my former minor comments, see my response under each response from the authors.
2) Line 60: change “environmental” to “the environment”
Response: Thanks for highlighting this point. It has been corrected in revised manuscript. Line number 63
Reviewer: Add “the” before “environmental”.
3) Line 62: chemical insensitivity is one type of resistance. There are other types of insecticide resistance.
4) Check the IRAC webpage: http://iraconline.org/about/resistance/mechanisms/. Delete “due to loss of chemical insensitivity” or change accordingly.
Response: Thank you for mentioning this point. It has been replaced with “due to development of insecticide”. Line number 65
Reviewer: (Line 64) I am not sure what do you mean by “Pesticide resistance is a long standing challenge… due to development of pesticide”. The sentence is redundant. Delete “due to development of pesticide” or explain using other words.
15) Line 224: Add “(fig. 1)” after “density”.
Response: Thanks for this technical suggestion. Corrected in revised manuscript. Line number 230
Reviewer: Not done. Line 232.
33) Line 547: I have not checked the author guidelines for references but usually book titles are preceded by “In:”
Response: Thanks for this critical highlight. Corrected in revised manuscrpt. Line number 583
Reviewer: (Line 605) name of book chapter between pages 275-342 is missing.
34) Line 549: Hyphen missing between “density” and “dependent”
Response: Thanks for this critical highlight. Corrected in revised manuscrpt. Line number 585
Reviewer: This was not done!
- Reviewer: NEW MINOR COMMENTS:
Line 177-184 (all new sentences in red): These sentences sound too casual, not formal.
Consider the following construction:
“Consumed preys were not replaced in our experiment and, therefore, Rogers’s Random Attack Model was used. This equation is appropriate for modelling predation or parasitism that causes a significant reduction of prey or host densities. In addition, our experimental data showed a type II functional response, which is appropriately analyzed by the random predator equation.”
I would not mention “Disk Equation”, otherwise you need to add references about it.
Line 206: add “is” between “temperature” and “non-significant”. Do the same for table 2.
Lines 274-294: Paragraphs are the same as the two previous one. Delete.
Line 320: there are two “including” on the same sentence that do not sound nice. Replace the second “including” by “, such as”.
Line 329: add a comma before “and”.
Line 345: this sentence is not correct. Add “the” before “highest” and “at 35oC” after “female”.
Line 363: the new sentence is basically repeating the sentence in line 361-363 (“The handling time for adult predators…”) but with more details. The addition of details is good but combine both these sentences into one and adjust the paragraph accordingly.
Line 365: “Seko and Miura”, not just “Seko”.
Line 366: “they”, not “he”. There are two authors in that reference.
Line 389: By “rich”, do you mean “abundant” or “nutritious”? Change accordingly.
Line 563-564: The author is “Hu, G.”, not “Guanfang, H.” Volume number is wrong; It is 14, not 4. Pages are missing; They are 180-185.
Lines 565-566: Scientific names in italic.
Line 619: Journal name should not be all in capital letters.
Table 1, Female, 15oC: Replace “d/8” by “d/*”. I think that is what the authors meant to be?
Author Response
Dear Editor and Reviewer,
On behalf of all authors, I am thankful for valuable comments and suggestions on the manuscript insects-865851, entitled "Temperature dependent functional response of Harmonia axyridis (Coleoptera: Coccinellidae) on the eggs of Spodoptera litura (Lepidoptera: Noctuidae) in laboratory". These comments are valuable and helpful for revising and improving the manuscript, as well as this is an important guidance for our future research.
Reply to the editor
Thanks. We addressed all the questions and changes in revised manuscript. And sorry we could not mention correct line numbers in response letter in previous revision. It happened due to copy paste problem and I think one reviewer review our file in PDF format and other reviewer in word. To overcome this problem, we are attaching two files. One is in word format with track changes and same file with accepted track changes in word format to overcome repetion problem., so that reviewer can see that there is no repetition. Because in PDF file that was sent to us for further revision, two paragraphs were two times but they were not present in Word file.
Secondly, We improved English and removed grammatical mistakes and made some changes in discussion for coherence as suggested by one reviewer with track changes.
If you think, this manuscript needs some changes more then let us know. Thanks Again!
Instructios for Reviewer: Please follow the word file for review with track changes not the PDF version.
------------------------------------------------------------
We replaced order of Acyrthosiphon pisum from Homoptera to Hemiptera. We made this changes in Materials and Methods. LINE NO 96
Now the order for this pest has changed. The reference is attached below:
Ximenez-Embuna, M.G. et al. (2014). Seasonal, spatial and diel partitioning of Acyrthosiphon pisum (Hemiptera: Aphididae) predators and predation in alfalfa fields. Biological Control 69, 1-7.
Reviewer #2
General comments:
The manuscript was greatly improved and I just have a few more comments. Please be aware that some of my comments were not addressed, although the authors said they were, and I am posting them again. In addition, authors cited wrong line numbers to the great majority of additions and corrections that they made. Some of them were not even close to the right line number. It took me a while to find them all and it is very time consuming. Before resubmitting your revised manuscript and letter, the last thing the author should do is making all line numbers are correct in their response letter.
Response: Thank you very much for appreciation. We are really sorry that we could not provide correct line numbers in the last revision. It happened due to copy paste issue. In this revision, we have very carefully checked page and line numbers, and all your comments have been replied to the best of our ability and understanding. Seconly, we have gone through the entire manuscipt very carefully, improved the English and removed typos or grammatical errors. Hope you find this version of the manuscript OK! Please follow the word file for review with track changes not PDF.
Major Comments:
1) Data Analysis Section: why estimating all these parameters (i.e., instantaneous attack rate, handling time, total searching time) is important for a biocontrol program? How are these parameters in the decision-making of an IPM strategy?
Reviewer: This is a good response. I think the manuscript would gain if you adapted this response into the discussion section. Please consider this request. Also, be aware that there are several typos in your response, in case you decide to incorporate it into your manuscript.
Thanks for mentioning this important point. Functional response is an imperative criterion for accessing the efficiency of a predator on a given prey. Handling time and attack rate also called (search rate or searching time or rate of discovery or space clearance rate) are important parameters of the functional response. The attack rate determines the ability of a predator to catch prey in a given time and handling time indicates the time a predator spends to identify, subjugate, attack and consume a particular prey [1]. These parameters determines the foraging behavior of a predator and tell how much efficient is a predator against a given density of prey. As the foraging behavior increase the attack rate will also increase which leads to decrease in handling time. It means the predator will consume more prey and we can use it to control pest. The parameters of functional response help to explain relative changes in the efficacy of the predator with changing density of the prey, along the growth of the predator. For example, early in the season when the pest density is low, the predator efficiency is also expected to be low, based on the parameters of functional response. Similarly, as the predator grows through different instars, the functional response parameters also improve. Therefore, the capacity of the predator to regulate the pest population will improve. Therefore, the functional response parameters give an indication of the temporal variation in the efficacy of the predator to regulate the pest population.
Response: We have added this paragraph to the discussion.
We have read the paragraph carefully and removed spelling and grammatical erros. Line Number (307-322)
2) Could you define what attack rate is? Is it number of eggs consumed/h? Attacks/h (what is the definition of attack here? When does the attack start and end? Attack to a batch of eggs? Sounds odd to me that a female would launch a 0.546 attack/h (at 35 oC). In this case, for instance, what is 0.546 attack?
On the same topic, how accurate are these coefficients compared to observed attack rates? Is there any reference for this?
Reviewer: A response to my second question about the accuracy of these coefficents compared to actual rates is missing. Any thoughts?
Response: In functional response experiments, attack rate coefficient describes ability of a predator to capture or grab the given prey. Rather being an observed value, attack rate is actually a derived or calculated value using standard equaltion or theoretical formula. For example, attack rate in our study was derived using Rogers random equation. Based on these calculations, we can assess which predator stage will be more efficient biological control agent. Also, this parameters are used to comapre the different stages of predator, so that we can make final decision which stage will be more beneficial for biological control.
3) Since the data is not normally distributed, have the authors considered using a Poisson glm? This distribution would be appropriate for counts (i.e., number of eggs consumed). I recommend trying this analysis because it seems odd that a mean prey consumption of, for example, females (65.78 +/- 17.21) is not different than of 1st instar (9.78 +/- 1.31), according to Table 1.
Response: Table 1 represents the results of the effect of tempearture and predator stage on the average prey consumption. We evluated the effect by non-parametric ANOVA given the data are not normal.
As suggested, we tried generalised linear model with poisson distribution and found that the data are over-dispersed (Sample variance >> Mean). Therefore, we fitted the negative binomial glm to the data followed by ANOVA and Tukey HSD. The results differ considerably from the earlier analysis. The results of the new analysis are given in Table 1.
Reviewer: Thank you for the pro-active action of analyzing the data using negative binomial once the poisson was not appropriate! Well done!
Response: Thank you very much for appreciation. We have added this information in materials and methods (Data Analysis) also. Line Number (162-164).
The data on average prey consumption were analyzed with generalized linear model assuming negative binomial distribution due to over-dispersion and group means were separated with Tukey’s HSD test (P<0.05).
Minor Comments:
1) Line 60: change “environmental” to “the environment”.
Add “the” before “environmental”.
Response: Done according to the reviewer suggestion. Line Number (63)
2) Line 62: chemical insensitivity is one type of resistance. There are other types of insecticide resistance.
3) Check the IRAC webpage: http://iraconline.org/about/resistance/mechanisms/. Delete “due to loss of chemical insensitivity” or change accordingly.
I am not sure what do you mean by “Pesticide resistance is a long standing challenge…due to development of pesticide”. The sentence is redundant. Delete “due to development of pesticide” or explain using other words.
Response: We Removed/Deleted these lines from manuscript. The new line starts from: Line Number (64)
4) Line 224: Add “(fig. 1)” after “density”.
Response: Done according to the reviewer suggestion. Line Number (231)
5) Line 547: I have not checked the author guidelines for references but usually book titles are preceded by “In:”
(Line 605) name of book chapter between pages 275-342 is missing.
Response: Name of the book chapter has been added. Line Number (595-596) Reference No 87
6) Line 549: Hyphen missing between “density” and “dependent”
Response: Done according to the reviewer suggestion. Line Number (597) Reference No 88
New Minor Comments:
1) Line 177-184: (all new sentences in red): These sentences sound too casual, not formal.
Consider the following construction:
“Consumed preys were not replaced in our experiments and, therefore, Rogers, s Random Attack Model was used. This equation is appropriate for modelling predation or parasitism that causes a significant reduction of prey or host densities. In addition, our experimental data showed a type II functional response, which is appropriately analyzed by the random predator equation”.
I would not mention “Disk Equation” , otherwise you need to add references about it.
Response: We removed these lines from materials and methods and added in results as suggested by reviewer 1. And I did not mention disk equation also. Line Number (222-224)
2) Line 206: add “is” between “temperature” and “non-significant”. Do the same for table 2.
Response: Thanks for this technical suggestion. We added according to the reviewer suggestion. Table 1 Line Number (203)
Table 2 Line Number (219)
3) Line 274-294: Paragraphs are the same as the two previous one. Delete.
Response: Thanks for this important highlight. This was actually a result of copy paste issue. We have removed repeating paragraph from the revised document. We also attached word format file with accepted track changes, so that you can review easily.
4) Line 320: there are two “including” on the same sentence that do not sound nice. Replace the second “including” by “such as”.
Response: Replaced according to the reviewer suggestion. Line Number (294)
5) Line 329: add a coma before “and”.
Response: Done as suggested by reviewer. Line Number (302)
6) Line 345: this sentence is not correct. Add “the” before “highest” and “at 35 °C” after “female”.
Response: We revised our discussion and also revised this sentence. Line Number (333-334)
7) Line 363: the new sentence is basically repeating the sentence in line 361-363 (“The handling time for adult predators…..”) but with more details. The addition of details is good but combine both these sentences into one adjust the paragraph accordingly.
Response: Thanks for this technical recommendation. We revised our discussion portion with track changes to make it coherent. Line Number (346-358)
8) Line 365: “Seko and Miura” , not just “Seko”
Response: Done as suggested by reviewer. Line Number (352)
9) Line 366: “they”, not “he”. There are two authors in that reference.
Response: We revised our discussion and deleted he. Line Number (353)
10) Line 389: By “rich”, do you mean “abundant” or “nutritious”? Change accordingly.
Response: Done as suggested by reviewer. Line Number (378)
11) Line 563-564: The author is “hu, G”, not “Guangfang, H.” Volume number is wrong; it is 14, not 4. Pages are missing. They are 180-185.
Response: Done as suggested by reviewer. Line Number (551-553), Reference No 68
12) Line 565-566: Scientific names in italic
Response: Done as suggested by reviewer. Line Number (554-555), Reference No 69
13) Line 619: Journal name should not be all in capital letters.
Response: Done as suggested by reviewer. Line Number (610) and Reference No 94
14) Table 1, Female, 15 °C: Replace d/8 by d/*. I think that is what the authors meant to be?
Response: Thanks for mentioning this important highlight. We revised it as suggested by reviewer. Table 1

Round 3
Reviewer 2 Report
Line 307: place the opening parenthesis before “also called”, not after.
Line 309: add a comma before “and”
Line 333: replace “Female” by “females: (low case “f” and plural)
Line 354: this construction is not correct. Delete period, add comma before “Whereas”.
Author Response
Dear Reviewer,
On behalf of all authors, I am thankful for valuable comments and suggestions on the manuscript insects-865851, entitled "Temperature dependent functional response of Harmonia axyridis (Coleoptera: Coccinellidae) on the eggs of Spodoptera litura (Lepidoptera: Noctuidae) in laboratory". These comments are valuable and helpful for revising and improving the manuscript, and we have checked and revised all comments and suggestions. Thanks again!
Reviewer 2 Round 3
Line 307: Place the opening parenthesis before “also called” , not after.
Response: It has been revised as suggested by reviewer. Line Number (307)
Line 309: Add a coma before “and”
Response: It has been revised as suggested by reviewer. Line Number (309)
Line 333: Replace “Female” by “females:” (low case “f” and plural)
Response: Thanks for mentioning this important point. Revised in manuscript. Line Number (333)
Line 354: This construction is not correct. Delete period, add coma before “Whereas”.
Response: It has been revised as suggested by reviewer. Line Number (354)
Kind Regards,
Xingmiao Zhou and co-authors